# Generalized Method of Mathematical Prototyping of Energy Processes for Digital Twins Development

**Sergey Khalyutin** [1,*], **Igor Starostin** [1] **and Irina Agafonkina** [2]

1   Department of Electrical Engineering and Aviation Electrical Equipment, Moscow State Technical University of Civil Aviation (MSTU CA), 125993 Moscow, Russia
2   Department of Chemical Engineering, Ariel University, Ariel 4070000, Israel
*   Correspondence: s.khalutin@mstuca.aero; Tel.: +7-(903)-549-84-01

**Abstract:** The use of digital twins in smart power systems at the stages of the life cycle is promising. The dynamics of such systems (smart energy renewable sources, smart energy hydrogen systems, etc.), are determined mainly by the physical and chemical processes occurring inside the systems. The basis for developing digital twins is reliable mathematical models of the systems. In the present paper, the authors present a method of energy processes mathematical prototyping—an overall approach to modeling processes of various physical and chemical natures based on modern non-equilibrium thermodynamics, mechanics, and electrodynamics. Controlled parameters are connected with measured ones by developing a theoretically correct system of process dynamics equations with accuracy up to the experimentally studied properties of substances and processes. Subsequent transformation into particular mathematical models of a specific class of systems makes this approach widely applicable. The properties of substances and processes are given in the form of functional dependencies on the state of the system up to experimentally determined constant coefficients. The authors consider algorithms for identifying the constant coefficients of the functions of substances and processes properties, which complement the proposed unified approach of designing models of various physical and chemical nature systems.

**Keywords:** mathematical prototyping method; energy processes; systems identification; symbolic regression; digital twins

## 1. Introduction

Utilization of renewable energy sources is primarily associated with the requirements of controlling a distributed system with unstable parameters and implies the need to automate the management of such systems. The complexity of the system's management appears from the flow of nonlinear physical and chemical processes in the system, which significantly depend not only on operating modes but also on external conditions (such as ambient temperature, wind, and solar insolation). Nowadays, promising solutions are smart control systems based on the creation and implementation of digital twins, in which the parameters of the objects are monitored, and their mathematical models are reconstructed. The basis for digital twin development is reliable mathematical models of the systems.

For the functioning of digital twins, it is necessary to use mathematical models in the form of dependencies of the controlled parameters on the measured ones. The measured parameters are input to the model, and the controlled parameters are returned at the output, according to which practical decisions are made [1–9]. In particular, probabilistic characteristics can be obtained at the output, for example, in problems related to the reliability and safety of technical systems [2,3,9].

Methods of identification theory [10,11], methods of machine learning [12–20], including deep learning based on neural networks [14], and symbolic regression [15–21] are often used to obtain mathematical models of renewable energy systems (RES). However, all

these modeling methods belong to the class of simulation models that do not consider actual physical and chemical processes and therefore do not guarantee their correctness in the entire range of operating conditions. To obtain correct (not contradicting the general physical laws) models of systems, it is necessary to set some restrictions arising from the general physical laws and the system functioning mechanism [10–23].

Correct mathematical models are formed based on physical laws and the study of the characteristics of the object of study: the structure of the object and the physical and chemical processes occurring in it. That is why the approach based on modeling the dynamics of physical and chemical processes in the system has found wide practical application, in contrast to the simulation approach, which considers the system as a "black box" [4,8–10,22,23]. Another aspect of resorting to this mathematical model approach is interpretability and explainability for subsequent decision-making.

In the general case, the physical laws, based on which the system model is built, are conservation laws or the connection between the causes of processes (internal disturbances, internal forces) and the rates of these processes (flows) [10,24–39]. This makes it possible to set a class of models for any RES in the form of a connection between internal forces and flows, as well as a connection between flows and the rates of change in its state coordinates based on conservation laws [10,24–39]. These models are generally built up to the experimentally determined properties of substances and processes, which generally depend on the state of the system [24–39]. Thus, the model of the particular RES instance should be a system of equations: differential equations for the dynamics of physical and chemical processes, as well as equations for measured and controlled parameters.

The features of building a system model described above make it possible to develop a unified approach to build models of systems of various physical and chemical nature [39,40], based on mechanics [29,30], including continuum mechanics [28,31], electrodynamics [28,32], the theory of electric and magnetic circuits [32], on modern non-equilibrium thermodynamics [33–38] and incorporating methods of identification theory, machine learning methods, including deep learning based on neural networks, and on symbolic regression [15–21]. Such models take measured parameters as input and return controlled parameters that have practical value as output [1–9]. This work is devoted to the development of this approach.

## 2. Description of the Energy Processes Mathematical Prototyping Method

As noted above, the general fundamental laws are conservation laws and the link between system internal disturbances and the speed of physical and chemical processes inside the system [10,24–39]. The main conservation law in mechanical systems is the momentum conservation law [28,30,31]; in the theory of electric and magnetic circuits, as well as in electrodynamics—the electric charge and magnetic flux [28,32] conservation law; in modern non-equilibrium thermodynamics—the energy conservation law (the first law of thermodynamics), the mass and stoichiometric ratios conservation law [28,31,33–38]. The energy conservation law is a general physical conservation law [28,30–38].

From the point of view of modern non-equilibrium thermodynamics, the cause and necessary condition for the occurrence of physicochemical processes in an arbitrary system are thermodynamic forces [33,35–37], which are internal disturbances [10,24–29,33,35–37]. Examples of thermodynamic forces are [33,35–37]: normalized temperature difference (reciprocal temperature difference), chemical potential difference, and chemical affinity. In mechanics, internal disturbances are the difference in velocities, which causes the friction force (momentum flux due to friction), as well as potential forces [10,28–31]. In electrodynamics, the theory of electrical and magnetic circuits, such internal disturbances are the difference in electrical potentials that carry an electric charge, and currents through inductive elements that create magnetic fluxes [10,28,29,32].

Flows in physical and chemical systems are the rates of physical and chemical processes [33,35–37]: in mechanics—mechanical forces (due to Newton's second law—momentum flows) and speeds [10,28–31]; in electrodynamics, theories of elec-

tric and magnetic circuits—electric currents that carry an electric charge and EMF of electromagnetic induction [10,28,29,32]. Examples of flows in modern nonequilibrium thermodynamics are [33,35–37]: heat flow, substance flow (diffusion flow, phase transition rate), and chemical reaction rate. These internal perturbations do not unambiguously determine the flows, the flows are also determined by the properties of the systems that do not depend on the perturbations. [10,24–39].

In the general case of processes of different physical and chemical nature, the authors proposed a unified approach to their description within the framework of mechanics [29,30], including continuum mechanics [28,31], electrodynamics [28,32], the theory of electrical and magnetic circuits [32], modern nonequilibrium thermodynamics [33–38]—it is a method of mathematical prototyping that does not contradict the conservation laws and the second law of thermodynamics [39]. The equations of this method can be written as [39]:

$$\frac{d\mathbf{x}(t)}{dt} = \mathbf{B}(\mathbf{x}(t), \mathbf{U}(t)) \frac{\delta \Delta \mathbf{x}(t)}{dt} + \frac{d\mathbf{x}^*(t)}{dt}, \ \frac{\delta \Delta \mathbf{x}(t)}{dt} = \mathbf{A}(\mathbf{x}(t), \mathbf{U}(t)) \cdot \Delta \mathbf{F}(\mathbf{x}(t), \mathbf{U}(t)), \quad (1)$$

$$\Delta \mathbf{F}(\mathbf{x}, \mathbf{U}) = \mathbf{B}^T(\mathbf{x}, \mathbf{U}) \cdot \mathbf{F}(\mathbf{x}, \mathbf{U}), \ \mathbf{F}(\mathbf{x}, \mathbf{U}) = -\nabla_{\mathbf{x}} W(\mathbf{x}, \mathbf{U}), \quad (2)$$

$$\mathbf{y}(t) = \mathbf{g}_{\mathbf{y}}(\mathbf{x}(t), \mathbf{U}(t)), \ \mathbf{z}(t) = \mathbf{g}_{\mathbf{z}}(\mathbf{x}(t), \mathbf{U}(t)), \quad (3)$$

where **x**—the coordinates of the system state; Δ**x**—coordinates of processes in the system; $\delta \Delta \mathbf{x}(t)/dt$—speed of physical and chemical processes in the system; $\mathbf{B}(\mathbf{x}, \mathbf{U})$—system topology matrix; $d\mathbf{x}^*(t)/dt$ is the component of the change rate of the system state parameters, due to its interaction with external systems (it also additively includes a random component of these external flows [36–38]); $\Delta \mathbf{F}(\mathbf{x}, \mathbf{U})$—dynamic forces (internal disturbances), which are the cause and necessary condition for the flow of physical and chemical processes in the system; $\mathbf{F}(\mathbf{x}, \mathbf{U})$—partial derivatives of the free energy $W(\mathbf{x}, \mathbf{U})$ by the coordinates of the state **x**, taken with the sign "-"; $\mathbf{A}(\mathbf{x}, \mathbf{U})$—positively defined (in particular cases, if there is inertia in the system, non-degenerated, non-negative defined) dissipative matrix; **U**—system parameters that do not change as a result of the processes in the system, but change as a result of external influences; **y**—measured parameters of the system; **z**—controlled parameters of the system. The measured and controlled parameters of the system (**y** and **z**) can be both a function of the system state and functionals of the system dynamics (dynamics **x**(*t*) and **U**(*t*)). To implement the system of Equations (1)–(3) in numerical form, it is necessary to have [39]:

- Topology matrix $\mathbf{B}(\mathbf{x}, \mathbf{U})$;
- The expression for the free energy $W(\mathbf{x}, \mathbf{U})$, expressed in terms of **x**, or its partial derivatives by the state coordinates **x**, taken with the "-" sign $\mathbf{F}(\mathbf{x}, \mathbf{U})$, satisfying the total differential condition;
- Positively defined dissipative matrix $\mathbf{A}(\mathbf{x}, \mathbf{U})$;
- Functions $\mathbf{g}_{\mathbf{y}}(\mathbf{x}, \mathbf{U})$ and $\mathbf{g}_{\mathbf{z}}(\mathbf{x}, \mathbf{U})$, obtained from the definition of the measured **y** and controlled **z** parameters of the system, respectively.

If we denote a part of the parameters **y** and **z** as $\bar{\mathbf{y}}$ and $\bar{\mathbf{z}}$, which are functions of the state **x**:

$$\bar{\mathbf{y}} = \bar{\mathbf{g}}_{\mathbf{y}}(\mathbf{x}, \mathbf{U}), \ \bar{\mathbf{z}} = \bar{\mathbf{g}}_{\mathbf{z}}(\mathbf{x}, \mathbf{U}), \quad (4)$$

then if the right-hand sides of (1) and (2) directly depend on $\bar{\mathbf{y}}$ and $\bar{\mathbf{z}}$, we represent the quantities included in (1) and (2) as compositions of functions [38]:

$$\mathbf{B}(\mathbf{x}, \mathbf{U}) \equiv \mathbf{B}(\bar{\mathbf{y}}, \bar{\mathbf{z}}, \mathbf{U}) \equiv \mathbf{B}(\bar{\mathbf{g}}_{\mathbf{y}}(\mathbf{x}, \mathbf{U}), \bar{\mathbf{g}}_{\mathbf{z}}(\mathbf{x}, \mathbf{U}), \mathbf{U}) \quad (5)$$

$$W(\mathbf{x}, \mathbf{U}) \equiv W(\bar{\mathbf{y}}, \bar{\mathbf{z}}, \mathbf{U}) \equiv W(\bar{\mathbf{g}}_{\mathbf{y}}(\mathbf{x}, \mathbf{U}), \bar{\mathbf{g}}_{\mathbf{z}}(\mathbf{x}, \mathbf{U}), \mathbf{U}) \quad (6)$$

$$\mathbf{F}(\mathbf{x}, \mathbf{U}) \equiv \mathbf{F}(\bar{\mathbf{y}}, \bar{\mathbf{z}}, \mathbf{U}) \equiv \mathbf{F}(\bar{\mathbf{g}}_{\mathbf{y}}(\mathbf{x}, \mathbf{U}), \bar{\mathbf{g}}_{\mathbf{z}}(\mathbf{x}, \mathbf{U}), \mathbf{U}) \quad (7)$$

$$\mathbf{A}(\mathbf{x}, \mathbf{U}) \equiv \mathbf{A}(\overline{\mathbf{y}}, \overline{\mathbf{z}}, \mathbf{U}) \equiv \mathbf{A}(\overline{\mathbf{g}}_{\mathbf{y}}(\mathbf{x}, \mathbf{U}), \overline{\mathbf{g}}_{\mathbf{z}}(\mathbf{x}, \mathbf{U}), \mathbf{U}) \tag{8}$$

$$\tilde{\mathbf{y}}(t) = \tilde{\mathbf{g}}_{\mathbf{y}}(\mathbf{x}(t), \mathbf{U}(t)) \equiv \tilde{\mathbf{g}}_{\mathbf{y}}(\overline{\mathbf{y}}(t), \overline{\mathbf{z}}(t), \mathbf{U}(t)) \equiv \tilde{\mathbf{g}}_{\mathbf{y}}(\overline{\mathbf{g}}_{\mathbf{y}}(\mathbf{x}(t), \mathbf{U}(t)), \overline{\mathbf{g}}_{\mathbf{z}}(\mathbf{x}(t), \mathbf{U}(t)), \mathbf{U}(t)) \tag{9}$$

$$\tilde{\mathbf{z}}(t) = \tilde{\mathbf{g}}_{\mathbf{z}}(\mathbf{x}(t), \mathbf{U}(t)) \equiv \tilde{\mathbf{g}}_{\mathbf{z}}(\overline{\mathbf{y}}(t), \overline{\mathbf{z}}(t), \mathbf{U}(t)) \equiv \tilde{\mathbf{g}}_{\mathbf{z}}(\overline{\mathbf{g}}_{\mathbf{y}}(\mathbf{x}(t), \mathbf{U}(t)), \overline{\mathbf{g}}_{\mathbf{z}}(\mathbf{x}(t), \mathbf{U}(t)), \mathbf{U}(t)) \tag{10}$$

where $\tilde{\mathbf{y}}$ and $\tilde{\mathbf{z}}$ are another part of the parameters $\mathbf{y}$ and $\mathbf{z}$, respectively:

$$\mathbf{y} = \left( \overline{\mathbf{y}}^T \quad \tilde{\mathbf{y}}^T \right)^T, \ \mathbf{z} = \left( \overline{\mathbf{z}}^T \quad \tilde{\mathbf{z}}^T \right)^T. \tag{11}$$

In the vast majority of cases, the absolute values of the coordinates of the state $\mathbf{x}$ are not determined (for example, the internal energy [38,41]), so, as can be seen from (4), it is advisable to set the quantities $\overline{\mathbf{y}}$ and $\overline{\mathbf{z}}$ in differential form [38]:

$$d\overline{\mathbf{y}} = \mathbf{C}_{\mathbf{y},\mathbf{x}}(\overline{\mathbf{y}}, \overline{\mathbf{z}}, \mathbf{U})d\mathbf{x} + \mathbf{C}_{\mathbf{y},\mathbf{U}}(\overline{\mathbf{y}}, \overline{\mathbf{z}}, \mathbf{U})d\mathbf{U}, \ d\overline{\mathbf{z}} = \mathbf{C}_{\mathbf{z},\mathbf{x}}(\overline{\mathbf{y}}, \overline{\mathbf{z}}, \mathbf{U})d\mathbf{x} + \mathbf{C}_{\mathbf{z},\mathbf{U}}(\overline{\mathbf{y}}, \overline{\mathbf{z}}, \mathbf{U})d\mathbf{U}, \tag{12}$$

where $\mathbf{C}_{\mathbf{y},\mathbf{x}}(\overline{\mathbf{y}}, \overline{\mathbf{z}}, \mathbf{U}), \mathbf{C}_{\mathbf{y},\mathbf{U}}(\overline{\mathbf{y}}, \overline{\mathbf{z}}, \mathbf{U}), \mathbf{C}_{\mathbf{z},\mathbf{x}}(\overline{\mathbf{y}}, \overline{\mathbf{z}}, \mathbf{U}), \mathbf{C}_{\mathbf{z},\mathbf{U}}(\overline{\mathbf{y}}, \overline{\mathbf{z}}, \mathbf{U})$ are the Jacobians of the right-hand sides of (4) with respect to $\mathbf{x}$ and $\mathbf{U}$. Hence, based on (1) and (12) we obtain the following expressions [38]:

$$\frac{d\overline{\mathbf{y}}(t)}{dt} = \tilde{\mathbf{B}}_{\mathbf{y}}(\overline{\mathbf{y}}(t), \overline{\mathbf{z}}(t), \mathbf{U}(t))\frac{\delta\Delta\mathbf{x}(t)}{dt} + \frac{d\overline{\mathbf{y}}^*(t)}{dt}, \ \frac{d\overline{\mathbf{z}}(t)}{dt} = \tilde{\mathbf{B}}_{\mathbf{z}}(\overline{\mathbf{y}}(t), \overline{\mathbf{z}}(t), \mathbf{U}(t))\frac{\delta\Delta\mathbf{x}(t)}{dt} + \frac{d\overline{\mathbf{z}}^*(t)}{dt}, \tag{13}$$

where [38]:

$$\frac{d\overline{\mathbf{y}}^*(t)}{dt} = \mathbf{C}_{\mathbf{y},\mathbf{x}}(\overline{\mathbf{y}}(t), \overline{\mathbf{z}}(t), \mathbf{U}(t))\frac{d\mathbf{x}^*(t)}{dt} + \mathbf{C}_{\mathbf{y},\mathbf{U}}(\overline{\mathbf{y}}(t), \overline{\mathbf{z}}(t), \mathbf{U}(t))\frac{d\mathbf{U}(t)}{dt}, \tag{14}$$

$$\frac{d\overline{\mathbf{z}}^*(t)}{dt} = \mathbf{C}_{\mathbf{z},\mathbf{x}}(\overline{\mathbf{y}}(t), \overline{\mathbf{z}}(t), \mathbf{U}(t))\frac{d\mathbf{x}^*(t)}{dt} + \mathbf{C}_{\mathbf{z},\mathbf{U}}(\overline{\mathbf{y}}(t), \overline{\mathbf{z}}(t), \mathbf{U}(t))\frac{d\mathbf{U}(t)}{dt}, \tag{15}$$

$$\tilde{\mathbf{B}}_{\mathbf{y}}(\overline{\mathbf{y}}, \overline{\mathbf{z}}, \mathbf{U}) = \mathbf{C}_{\mathbf{y},\mathbf{x}}(\overline{\mathbf{y}}, \overline{\mathbf{z}}, \mathbf{U}) \cdot \mathbf{B}(\overline{\mathbf{y}}, \overline{\mathbf{z}}, \mathbf{U}), \ \tilde{\mathbf{B}}_{\mathbf{z}}(\overline{\mathbf{y}}, \overline{\mathbf{z}}, \mathbf{U}) = \mathbf{C}_{\mathbf{z},\mathbf{x}}(\overline{\mathbf{y}}, \overline{\mathbf{z}}, \mathbf{U}) \cdot \mathbf{B}(\overline{\mathbf{y}}, \overline{\mathbf{z}}, \mathbf{U}). \tag{16}$$

By virtue of (5)–(10), system (1)–(3) takes the form [38]:

$$\frac{\delta\Delta\mathbf{x}(t)}{dt} = \mathbf{A}(\overline{\mathbf{y}}(t), \overline{\mathbf{z}}(t), \mathbf{U}(t)) \cdot \Delta\mathbf{F}(\overline{\mathbf{y}}(t), \overline{\mathbf{z}}(t), \mathbf{U}(t)), \ \Delta\mathbf{F}(\overline{\mathbf{y}}, \overline{\mathbf{z}}, \mathbf{U}) = \mathbf{B}^T(\overline{\mathbf{y}}, \overline{\mathbf{z}}, \mathbf{U}) \cdot \mathbf{F}(\overline{\mathbf{y}}, \overline{\mathbf{z}}, \mathbf{U}), \tag{17}$$

$$\tilde{\mathbf{y}}(t) = \tilde{\mathbf{g}}_{\mathbf{y}}(\overline{\mathbf{y}}(t), \overline{\mathbf{z}}(t), \mathbf{U}(t)), \ \tilde{\mathbf{z}}(t) = \tilde{\mathbf{g}}_{\mathbf{z}}(\overline{\mathbf{y}}(t), \overline{\mathbf{z}}(t), \mathbf{U}(t)). \tag{18}$$

The system of Equations (13)–(18) is more convenient for practical applications [38].

Thus, in order to obtain the transformed equations of the mathematical prototyping method (13)–(18), it is necessary to obtain functional dependences of the system state from the experimental data for substances and processes properties and system topology [38,39]:

- Topology matrix $\mathbf{B}(\overline{\mathbf{y}}, \overline{\mathbf{z}}, \mathbf{U})$;

- Positively defined dissipative matrix $\mathbf{A}(\overline{\mathbf{y}}, \overline{\mathbf{z}}, \mathbf{U})$;

- Jacobi matrices $\mathbf{C}_{\mathbf{y},\mathbf{x}}(\overline{\mathbf{y}}, \overline{\mathbf{z}}, \mathbf{U})$, $\mathbf{C}_{\mathbf{y},\mathbf{U}}(\overline{\mathbf{y}}, \overline{\mathbf{z}}, \mathbf{U})$ and $\mathbf{C}_{\mathbf{z},\mathbf{x}}(\overline{\mathbf{y}}, \overline{\mathbf{z}}, \mathbf{U})$, $\mathbf{C}_{\mathbf{z},\mathbf{U}}(\overline{\mathbf{y}}, \overline{\mathbf{z}}, \mathbf{U})$ of measured $\overline{\mathbf{y}}$ and controlled $\overline{\mathbf{z}}$ parameters of the system respectively, satisfying the conditions of the total differential [38]:

$$\sum_{k=1}^{n_{\overline{\mathbf{y}}}} \frac{\partial \mathbf{C}_{\mathbf{y},\mathbf{x},i}(\overline{\mathbf{y}},\overline{\mathbf{z}},\mathbf{U})}{\partial \overline{y}_k} C_{\mathbf{y},\mathbf{x},k,j}(\overline{\mathbf{y}},\overline{\mathbf{z}},\mathbf{U}) + \sum_{k=1}^{n_{\overline{\mathbf{z}}}} \frac{\partial \mathbf{C}_{\mathbf{z},\mathbf{x},i}(\overline{\mathbf{y}},\overline{\mathbf{z}},\mathbf{U})}{\partial \overline{z}_k} C_{\mathbf{z},\mathbf{x},k,j}(\overline{\mathbf{y}},\overline{\mathbf{z}},\mathbf{U}) \equiv$$

$$\equiv \sum_{k=1}^{n_{\overline{\mathbf{y}}}} \frac{\partial \mathbf{C}_{\mathbf{y},\mathbf{x},j}(\overline{\mathbf{y}},\overline{\mathbf{z}},\mathbf{U})}{\partial \overline{y}_k} C_{\mathbf{y},\mathbf{x},k,i}(\overline{\mathbf{y}},\overline{\mathbf{z}},\mathbf{U}) + \sum_{k=1}^{n_{\overline{\mathbf{z}}}} \frac{\partial \mathbf{C}_{\mathbf{z},\mathbf{x},j}(\overline{\mathbf{y}},\overline{\mathbf{z}},\mathbf{U})}{\partial \overline{z}_k} C_{\mathbf{z},\mathbf{x},k,i}(\overline{\mathbf{y}},\overline{\mathbf{z}},\mathbf{U}),$$

$$j = 1, i-1, \ i = 2, n_{\mathbf{x}}, \ n_{\mathbf{x}} = \dim(\mathbf{x}), \ n_{\overline{\mathbf{y}}} = \dim(\overline{\mathbf{y}}), \ n_{\overline{\mathbf{z}}} = \dim(\overline{\mathbf{z}}), \ n_{\mathbf{x}} = n_{\overline{\mathbf{y}}} + n_{\overline{\mathbf{z}}},$$

(19)

$$\sum_{k=1}^{n_{\overline{\mathbf{y}}}} \frac{\partial \mathbf{C}_{\mathbf{y},\mathbf{x},i}(\overline{\mathbf{y}},\overline{\mathbf{z}},\mathbf{U})}{\partial \overline{y}_k} C_{\mathbf{y},\mathbf{U},k,j}(\overline{\mathbf{y}},\overline{\mathbf{z}},\mathbf{U}) + \sum_{k=1}^{n_{\overline{\mathbf{z}}}} \frac{\partial \mathbf{C}_{\mathbf{z},\mathbf{x},i}(\overline{\mathbf{y}},\overline{\mathbf{z}},\mathbf{U})}{\partial \overline{z}_k} C_{\mathbf{z},\mathbf{U},k,j}(\overline{\mathbf{y}},\overline{\mathbf{z}},\mathbf{U}) + \frac{\partial \mathbf{C}_{\mathbf{y},\mathbf{x},i}(\overline{\mathbf{y}},\overline{\mathbf{z}},\mathbf{U})}{\partial U_j} \equiv$$

$$\equiv \sum_{k=1}^{n_{\overline{\mathbf{y}}}} \frac{\partial \mathbf{C}_{\mathbf{y},\mathbf{U},j}(\overline{\mathbf{y}},\overline{\mathbf{z}},\mathbf{U})}{\partial \overline{y}_k} C_{\mathbf{y},\mathbf{x},k,i}(\overline{\mathbf{y}},\overline{\mathbf{z}},\mathbf{U}) + \sum_{k=1}^{n_{\overline{\mathbf{z}}}} \frac{\partial \mathbf{C}_{\mathbf{z},\mathbf{U},j}(\overline{\mathbf{y}},\overline{\mathbf{z}},\mathbf{U})}{\partial \overline{z}_k} C_{\mathbf{z},\mathbf{x},k,i}(\overline{\mathbf{y}},\overline{\mathbf{z}},\mathbf{U}), \ j = 1, n_{\mathbf{U}}, \ i = 1, n_{\mathbf{x}},$$

$$n_{\mathbf{U}} = \dim(\mathbf{U}),$$

(20)

$$\sum_{k=1}^{n_{\overline{\mathbf{y}}}} \frac{\partial \mathbf{C}_{\mathbf{y},\mathbf{U},i}(\overline{\mathbf{y}},\overline{\mathbf{z}},\mathbf{U})}{\partial \overline{y}_k} C_{\mathbf{y},\mathbf{U},k,j}(\overline{\mathbf{y}},\overline{\mathbf{z}},\mathbf{U}) + \sum_{k=1}^{n_{\overline{\mathbf{z}}}} \frac{\partial \mathbf{C}_{\mathbf{z},\mathbf{U},i}(\overline{\mathbf{y}},\overline{\mathbf{z}},\mathbf{U})}{\partial \overline{z}_k} C_{\mathbf{z},\mathbf{U},k,j}(\overline{\mathbf{y}},\overline{\mathbf{z}},\mathbf{U}) + \frac{\partial \mathbf{C}_{\mathbf{y},\mathbf{U},i}(\overline{\mathbf{y}},\overline{\mathbf{z}},\mathbf{U})}{\partial U_j} \equiv$$

$$\equiv \sum_{k=1}^{n_{\overline{\mathbf{y}}}} \frac{\partial \mathbf{C}_{\mathbf{y},\mathbf{U},j}(\overline{\mathbf{y}},\overline{\mathbf{z}},\mathbf{U})}{\partial \overline{y}_k} C_{\mathbf{y},\mathbf{U},k,i}(\overline{\mathbf{y}},\overline{\mathbf{z}},\mathbf{U}) + \sum_{k=1}^{n_{\overline{\mathbf{z}}}} \frac{\partial \mathbf{C}_{\mathbf{z},\mathbf{U},j}(\overline{\mathbf{y}},\overline{\mathbf{z}},\mathbf{U})}{\partial \overline{z}_k} C_{\mathbf{z},\mathbf{U},k,i}(\overline{\mathbf{y}},\overline{\mathbf{z}},\mathbf{U}) + \frac{\partial \mathbf{C}_{\mathbf{z},\mathbf{U},j}(\overline{\mathbf{y}},\overline{\mathbf{z}},\mathbf{U})}{\partial U_i},$$

$$j = 1, i-1, \ i = 2, n_{\mathbf{U}},$$

(21)

where:

$$\mathbf{C}_{\mathbf{y},\mathbf{x}}(\overline{\mathbf{y}},\overline{\mathbf{z}},\mathbf{U}) = \begin{pmatrix} \mathbf{C}_{\mathbf{y},\mathbf{x},1}(\overline{\mathbf{y}},\overline{\mathbf{z}},\mathbf{U}) & \cdots & \mathbf{C}_{\mathbf{y},\mathbf{x},n_{\mathbf{x}}}(\overline{\mathbf{y}},\overline{\mathbf{z}},\mathbf{U}) \end{pmatrix},$$

$$\mathbf{C}_{\mathbf{z},\mathbf{x}}(\overline{\mathbf{y}},\overline{\mathbf{z}},\mathbf{U}) = \begin{pmatrix} \mathbf{C}_{\mathbf{z},\mathbf{x},1}(\overline{\mathbf{y}},\overline{\mathbf{z}},\mathbf{U}) & \cdots & \mathbf{C}_{\mathbf{z},\mathbf{x},n_{\mathbf{x}}}(\overline{\mathbf{y}},\overline{\mathbf{z}},\mathbf{U}) \end{pmatrix},$$

$$\mathbf{C}_{\mathbf{y},\mathbf{U}}(\overline{\mathbf{y}},\overline{\mathbf{z}},\mathbf{U}) = \begin{pmatrix} \mathbf{C}_{\mathbf{y},\mathbf{U},1}(\overline{\mathbf{y}},\overline{\mathbf{z}},\mathbf{U}) & \cdots & \mathbf{C}_{\mathbf{y},\mathbf{U},n_{\mathbf{x}}}(\overline{\mathbf{y}},\overline{\mathbf{z}},\mathbf{U}) \end{pmatrix},$$

$$\mathbf{C}_{\mathbf{z},\mathbf{U}}(\overline{\mathbf{y}},\overline{\mathbf{z}},\mathbf{U}) = \begin{pmatrix} \mathbf{C}_{\mathbf{z},\mathbf{U},1}(\overline{\mathbf{y}},\overline{\mathbf{z}},\mathbf{U}) & \cdots & \mathbf{C}_{\mathbf{z},\mathbf{U},n_{\mathbf{x}}}(\overline{\mathbf{y}},\overline{\mathbf{z}},\mathbf{U}) \end{pmatrix},$$

$$\mathbf{C}_{\mathbf{y},\mathbf{x}}(\overline{\mathbf{y}},\overline{\mathbf{z}},\mathbf{U}) = \begin{pmatrix} C_{\mathbf{y},\mathbf{x},1,1}(\overline{\mathbf{y}},\overline{\mathbf{z}},\mathbf{U}) & \cdots & C_{\mathbf{y},\mathbf{x},1,n_{\mathbf{x}}}(\overline{\mathbf{y}},\overline{\mathbf{z}},\mathbf{U}) \\ \vdots & \cdots & \vdots \\ C_{\mathbf{y},\mathbf{x},n_{\overline{\mathbf{y}}},1}(\overline{\mathbf{y}},\overline{\mathbf{z}},\mathbf{U}) & \cdots & C_{\mathbf{y},\mathbf{x},n_{\overline{\mathbf{y}}},n_{\mathbf{x}}}(\overline{\mathbf{y}},\overline{\mathbf{z}},\mathbf{U}) \end{pmatrix},$$

$$\mathbf{C}_{\mathbf{z},\mathbf{x}}(\overline{\mathbf{y}},\overline{\mathbf{z}},\mathbf{U}) = \begin{pmatrix} C_{\mathbf{z},\mathbf{x},1,1}(\overline{\mathbf{y}},\overline{\mathbf{z}},\mathbf{U}) & \cdots & C_{\mathbf{z},\mathbf{x},1,n_{\mathbf{x}}}(\overline{\mathbf{y}},\overline{\mathbf{z}},\mathbf{U}) \\ \vdots & \cdots & \vdots \\ C_{\mathbf{z},\mathbf{x},n_{\overline{\mathbf{y}}},1}(\overline{\mathbf{y}},\overline{\mathbf{z}},\mathbf{U}) & \cdots & C_{\mathbf{z},\mathbf{x},n_{\overline{\mathbf{y}}},n_{\mathbf{x}}}(\overline{\mathbf{y}},\overline{\mathbf{z}},\mathbf{U}) \end{pmatrix},$$

$$\mathbf{C}_{\mathbf{y},\mathbf{U}}(\overline{\mathbf{y}},\overline{\mathbf{z}},\mathbf{U}) = \begin{pmatrix} C_{\mathbf{y},\mathbf{U},1,1}(\overline{\mathbf{y}},\overline{\mathbf{z}},\mathbf{U}) & \cdots & C_{\mathbf{y},\mathbf{U},1,n_{\mathbf{x}}}(\overline{\mathbf{y}},\overline{\mathbf{z}},\mathbf{U}) \\ \vdots & \cdots & \vdots \\ C_{\mathbf{y},\mathbf{U},n_{\overline{\mathbf{y}}},1}(\overline{\mathbf{y}},\overline{\mathbf{z}},\mathbf{U}) & \cdots & C_{\mathbf{y},\mathbf{U},n_{\overline{\mathbf{y}}},n_{\mathbf{x}}}(\overline{\mathbf{y}},\overline{\mathbf{z}},\mathbf{U}) \end{pmatrix},$$

$$\mathbf{C}_{\mathbf{z},\mathbf{U}}(\overline{\mathbf{y}},\overline{\mathbf{z}},\mathbf{U}) = \begin{pmatrix} C_{\mathbf{z},\mathbf{U},1,1}(\overline{\mathbf{y}},\overline{\mathbf{z}},\mathbf{U}) & \cdots & C_{\mathbf{z},\mathbf{U},1,n_{\mathbf{x}}}(\overline{\mathbf{y}},\overline{\mathbf{z}},\mathbf{U}) \\ \vdots & \cdots & \vdots \\ C_{\mathbf{z},\mathbf{U},n_{\overline{\mathbf{y}}},1}(\overline{\mathbf{y}},\overline{\mathbf{z}},\mathbf{U}) & \cdots & C_{\mathbf{z},\mathbf{U},n_{\overline{\mathbf{y}}},n_{\mathbf{x}}}(\overline{\mathbf{y}},\overline{\mathbf{z}},\mathbf{U}) \end{pmatrix};$$

- Partial derivatives $\mathbf{F}(\overline{\mathbf{y}}, \overline{\mathbf{z}}, \mathbf{U})$ by the coordinates of the state $\mathbf{x}$ of free energy $W(\mathbf{x}, \mathbf{U})$, satisfying the conditions of the total differential [38]:

$$
\sum_{k=1}^{n_{\overline{\mathbf{y}}}} \frac{\partial F_i(\overline{\mathbf{y}}, \overline{\mathbf{z}}, \mathbf{U})}{\partial \overline{y}_k} C_{\mathbf{y}, \mathbf{x}, k, j}(\overline{\mathbf{y}}, \overline{\mathbf{z}}, \mathbf{U}) + \sum_{k=1}^{n_{\overline{\mathbf{z}}}} \frac{\partial F_i(\overline{\mathbf{y}}, \overline{\mathbf{z}}, \mathbf{U})}{\partial \overline{z}_k} C_{\mathbf{z}, \mathbf{x}, k, j}(\overline{\mathbf{y}}, \overline{\mathbf{z}}, \mathbf{U}) \equiv
$$
$$
\equiv \sum_{k=1}^{n_{\overline{\mathbf{y}}}} \frac{\partial F_j(\overline{\mathbf{y}}, \overline{\mathbf{z}}, \mathbf{U})}{\partial \overline{y}_k} C_{\mathbf{y}, \mathbf{x}, k, i}(\overline{\mathbf{y}}, \overline{\mathbf{z}}, \mathbf{U}) + \sum_{k=1}^{n_{\overline{\mathbf{z}}}} \frac{\partial F_j(\overline{\mathbf{y}}, \overline{\mathbf{z}}, \mathbf{U})}{\partial \overline{z}_k} C_{\mathbf{z}, \mathbf{x}, k, i}(\overline{\mathbf{y}}, \overline{\mathbf{z}}, \mathbf{U}),
$$
$$
j = 1, i - 1, \ i = 2, n_{\mathbf{x}}; \tag{22}
$$

- Functions of measured $\tilde{\mathbf{g}}_{\mathbf{y}}(\overline{\mathbf{y}}, \overline{\mathbf{z}}, \mathbf{U})$ and controlled $\tilde{\mathbf{g}}_{\mathbf{z}}(\overline{\mathbf{y}}, \overline{\mathbf{z}}, \mathbf{U})$ parameters (which can be functionals of $\overline{\mathbf{y}}(t)$, $\overline{\mathbf{z}}(t)$, and $\mathbf{U}(t)$ dynamics).

Further, for the dissipative matrix, partial derivatives of the free energy, the Jacobi matrices of the measured and controlled parameters specified in differential form, the topology matrix, and functions for the remaining measured and controlled parameters, it is necessary to specify a class of analytical expressions up to constant coefficients. In this class, for any desired function, there must necessarily be an analytic expression that approximates the desired function with any given accuracy. Such classes, for example, are:

- Power polynomials, whose convergence is guaranteed by the Weierstrass theorem on the uniform approximation of functions by polynomials [21];
- Classes of inductively generating functions [15] obtained by symbolic regression methods; convergence, in this case, is confirmed by the universal approximation theorem (Cybenko's theorem) [42,43];
- Classes of interpolation expressions (linear, cubic splines, Lagrange interpolation polynomials, etc.) [44].

The values of constant coefficients, in the general case, are determined in such a way that the measured parameters $\mathbf{y}$ determined from (1)–(3) or from (13)–(18) coincide with the corresponding values of these parameters $\mathbf{y}^{(E)}$, obtained by direct measurement [10,11,37]:

$$
\mathbf{y}(t_j) = \mathbf{y}^{(E)}(t_j), \ j = 1, N_t, \tag{23}
$$

where $t_j$, $j = 1, N_t$—discrete moments of time; $N_t$—number of discrete moments of time. The coefficients of approximating analytical expressions for the properties of substances and processes that satisfy the relevant restrictions can be determined from experimental data using (13)–(18), (23), by minimizing the objective function $Q[\mathbf{y}(t)]$ [10,11,37]:

$$
Q[\mathbf{y}(t)] = \frac{1}{2} \sum_{i=1}^{n} \sum_{j=1}^{N_{t,i}} (\mathbf{y}_i(t_j) - \mathbf{y}_i^{(E)}(t_j))^T \tilde{\mathbf{L}}_i (\mathbf{y}_i(t_j) - \mathbf{y}_i^{(E)}(t_j)), \tag{24}
$$

where $\tilde{\mathbf{L}}_i$, $i = 1, n$ is a positively defined symmetric matrix; $N_{t,i}$, $i = 1, n$—the number of discrete moments of time in each $i$-th operation mode; $n$ is the number of modes in which experimental data are taken $\mathbf{y}_i^{(E)}(t_j)$, $j = 1, N_{t,i}$, $i = 1, n$ [10,11,37]. Minimization of the objective function $Q$, defined by virtue of (13)–(18), (23), (24) can be carried out in different ways [45], for example, in parts [45]. Hence, the analytical expressions for the state function of the dissipative matrix should be given in the form [37]:

$$
\mathbf{A}(\overline{\mathbf{y}}, \overline{\mathbf{z}}, \mathbf{U}) = \mathbf{A}^\circ(\breve{\mathbf{h}}_a(\overline{\mathbf{y}}, \overline{\mathbf{z}}, \mathbf{U}, \mathbf{p}_a)) + \sum_{j=0}^{m_a} \breve{\mathbf{A}}_j \left( \prod_{i=1}^{n_{\mathbf{x}}} \frac{(\breve{h}_{a,i}(\overline{\mathbf{y}}, \overline{\mathbf{z}}, \mathbf{U}, \mathbf{p}_a))^{n_{i,j}}}{n_{i,j}!} \right),
$$
$$
\breve{\mathbf{h}}_a(\overline{\mathbf{y}}, \overline{\mathbf{z}}, \mathbf{U}, \mathbf{p}) = ( \ \breve{h}_{a,1}(\overline{\mathbf{y}}, \overline{\mathbf{z}}, \mathbf{U}, \mathbf{p}_a) \ \cdots \ \breve{h}_{a,n_{\mathbf{x}}}(\overline{\mathbf{y}}, \overline{\mathbf{z}}, \mathbf{U}, \mathbf{p}_a) \ )^T, \tag{25}
$$

where $\mathbf{A}^{\circ}(\breve{\mathbf{h}}_a)$ is the positively defined (non-negative-defined in the case of inertia in the system) basic dissipative matrix; $\breve{h}_{a,i} = \breve{h}_{a,i}(\overline{\mathbf{y}}, \overline{\mathbf{z}}, \mathbf{U}, \mathbf{p}_a) > 0$, $j = 0, m_a$ are some variables determined by the state of the system; $m_a$ is the number of basic additive functions; $\breve{\mathbf{A}}_j$, $j = 0, m_a$ are constant non-negative defined matrices; $\mathbf{p}_a$ are the parameters by which, along with $\breve{\mathbf{A}}_j$, $j = 0, m_a$ the objective function is optimized $Q$. For independent components of other properties of substances and processes (partial derivatives of free energy $\mathbf{F}(\overline{\mathbf{y}}, \overline{\mathbf{z}}, \mathbf{U})$, Jacobi matrices $\mathbf{C}_{\mathbf{y},\mathbf{x}}(\overline{\mathbf{y}}, \overline{\mathbf{z}}, \mathbf{U})$, $\mathbf{C}_{\mathbf{y},\mathbf{U}}(\overline{\mathbf{y}}, \overline{\mathbf{z}}, \mathbf{U})$ and $\mathbf{C}_{\mathbf{z},\mathbf{x}}(\overline{\mathbf{y}}, \overline{\mathbf{z}}, \mathbf{U})$, $\mathbf{C}_{\mathbf{z},\mathbf{U}}(\overline{\mathbf{y}}, \overline{\mathbf{z}}, \mathbf{U})$ observed $\overline{\mathbf{y}}$ and controlled $\overline{\mathbf{z}}$ parameters, respectively, which are taken in differential form, topology matrices $\mathbf{B}(\overline{\mathbf{y}}, \overline{\mathbf{z}}, \mathbf{U})$), analytical expressions are given in the form [37]:

$$H(\overline{\mathbf{y}}, \overline{\mathbf{z}}, \mathbf{U}) = H^{\circ}(\breve{\mathbf{h}}(\overline{\mathbf{y}}, \overline{\mathbf{z}}, \mathbf{U}, \mathbf{p})) + \sum_{j=0}^{m} c_j \left( \prod_{i=1}^{n_{\mathbf{x}}} \frac{(\breve{h}_i(\overline{\mathbf{y}}, \overline{\mathbf{z}}, \mathbf{U}, \mathbf{p}))^{n_{i,j}}}{n_{i,j}!} \right),$$

$$\breve{\mathbf{h}}(\overline{\mathbf{y}}, \overline{\mathbf{z}}, \mathbf{U}, \mathbf{p}) = \left( \begin{array}{ccc} \breve{h}_1(\overline{\mathbf{y}}, \overline{\mathbf{z}}, \mathbf{U}, \mathbf{p}) & \cdots & \breve{h}_{n_{\mathbf{x}}}(\overline{\mathbf{y}}, \overline{\mathbf{z}}, \mathbf{U}, \mathbf{p}) \end{array} \right)^T,$$

$$(26)$$

where $H^{\circ}(\breve{\mathbf{h}})$ is the basic component; $c_j$ are constant coefficients, $j = 0, m$; $\breve{h}_i = \breve{h}_i(\overline{\mathbf{y}}, \overline{\mathbf{z}}, \mathbf{U}, \mathbf{p})$, $i = 1, m$-some variables determined by the state of the system; $\mathbf{p}$ are the parameters by which, along with $c_j$ the objective function is optimized $Q$.

A positively defined dissipative matrix $\mathbf{A}(\overline{\mathbf{y}}, \overline{\mathbf{z}}, \mathbf{U})$, in the case of physical and chemical processes, is a kinetic matrix constructed through its reversible and positive irreversible components [36,37], which are given in the form (26) [37]. The positivity of the irreversible components guarantees the positive definiteness of the kinetic matrix [36,37]. In the case of mechanics, system (1)–(3) transforms into the Hamiltonian equations [30,39], in this case, the dissipative matrix is constructed based on the friction coefficients and transfer functions [39]. The positivity of the friction coefficients, also given in the form (26), guarantees the non-negative definiteness of the dissipative matrix of mechanical systems (included in the Hamilton equations) [30,39]. In the case of electrodynamics, in terms of the transfer of electric charge through an anisotropic crystal, in terms of the Hall effect, it is also expedient to build a dissipative matrix through reversible and irreversible components [35,36]. In the case of electric and magnetic circuits, the dissipative matrix is built based on the resistances in the electric circuit and the way they are connected [39]. For physical and chemical processes, the existing model of the specific nature process can be converted to the form (1)–(3) or to the form (13)–(18), in this case the analytical expression of the kinetic matrix is constructed in accordance with (25) [36].

Thus, building a model of physical and chemical processes in the system is reduced to optimizing the objective function $Q[\mathbf{y}(\tau)]$ given in the form (13)–(18), (23)–(26), according to the constant parameters included in (25) and (26). Such a procedure, as it is easy to see from (13)–(18), (23)–(26), is reduced to solving a system of ordinary differential equations. Currently, in order to solve a system of ordinary differential equations, the following methods [46] can be used:

- Step methods based on calculating the values of a dynamic variable at subsequent time points from the previous values of this variable;
- Special methods based on the approximation of solutions to a system of differential equations by analytical expressions.

The main advantage of step methods is their universality for any system of ordinary differential equations because as the integration step tends to zero, the approximate solution uniformly converges to the exact one [46]. However, these methods have a disadvantage—the complexity of calculations [46]. Special methods based on specifying the solution by an analytic expression followed by the search for the constant parameters

of this expression are free from this shortcoming [46]. This approach is less labor intensive than step methods. However, to apply this approach, it is necessary to take into account the qualitative nature of the solutions to the system of ordinary differential equations [46]. As an approximate solution, one can take a biunique function of the general analytical solution of a simplified system of differential equations [46]. Moreover, in the case of an autonomous system of differential equations, such a solution will have the property of a group and be a solution of an autonomous system of differential equations [47]. Such an approximate solution can be composed with analytical solutions of local simplifications of the system of ordinary differential equations being solved [46] by taking a biunique function from this solution composed of pieces [46].

It should be noted that in the absence of external flows, the system tends to an equilibrium state, and from any initial state (due to the zeroth law of thermodynamics) [34,36,37]. In the case of the presence of external flows, the system can evolve either into a stable stationary state or into an oscillatory regime (self-oscillations, forced oscillations, dynamic chaos, etc.) [48,49]. Oscillations in the system and their occurrence in the considered case are explained by the tendency of the system to a stationary state, which can change as a result of feedback [49]. Local simplifications of the equation's parameters of the mathematical prototyping method can be piecewise constant or piecewise polynomial [50]. Moreover, these simplifications can be chosen by taking the piecewise constant dissipative matrix, balance matrix, quantities $\mathbf{U}$, and external flows [48,49]. In such local areas, the system tends to a stationary state, which can also change as a result of a change in the balance parameters (for example, the total mass of the system, the total energy of the system, the total momentum of the system, etc.) [49]. Yet upon transition to another region, the stationary state can also change as a result of changes in the properties of the system [49]. Thus, an oscillatory motion of the system arises [49]. The general analytical solution of the system of equations of the mathematical prototyping method is obtained by stitching (without the refinement of the stitching method) simplified analytical solutions in local areas [46], which, in the limit, as the size of the areas tends to zero, converges to the analytical solution of the main system of Equations (1) and (2) [46].

The simplified analytic solutions $\tilde{\mathbf{x}}(t) = \tilde{\mathbf{x}}_t(\tilde{\mathbf{x}}_0, t)$ have the group property [47]:

$$\tilde{\mathbf{x}}_0 = \tilde{\mathbf{x}}_t(\tilde{\mathbf{x}}_0, 0), \ \tilde{\mathbf{x}}_t(\tilde{\mathbf{x}}_t(\tilde{\mathbf{x}}_0, \tau), t) = \tilde{\mathbf{x}}_t(\tilde{\mathbf{x}}_0, t + \tau). \tag{27}$$

Approximate general solution of Equations (1) and (2) $\overline{\mathbf{x}}(t) = \overline{\mathbf{x}}_t(\overline{\mathbf{x}}_0, t_0, t)$, where $\overline{\mathbf{x}}_0 = \overline{\mathbf{x}}_t(\overline{\mathbf{x}}_0, t_0, t_0)$, considering (27) is represented as:

$$\overline{\mathbf{x}}_t(\overline{\mathbf{x}}_0, t_0, t) \equiv \tilde{\mathbf{x}}_t(\tilde{\mathbf{x}}_{i-1}^{(0)}, t) \Leftrightarrow t \in [t_{i-1} - t_0, t_i - t_0], \ \tilde{\mathbf{x}}_i^{(0)} = \tilde{\mathbf{x}}_t(\tilde{\mathbf{x}}_{i-1}^{(0)}, t_i), \ \tilde{\mathbf{x}}_0^{(0)} = \overline{\mathbf{x}}_0, \ i = 1, \infty, \tag{28}$$

where $\tilde{\mathbf{x}}_i^{(0)}$, $i = 1, \infty$ additively includes a random component.

If, in the space of system state coordinates, we find such a basis under which the expressions for the parameters of the mathematical prototyping method equations system (functions of the dissipative matrix, topology matrix, the scalar function of free energy or its partial derivatives) are simplified. In the new space of system state coordinates, the number of local areas into which the entire space should be divided decreases.

To transform the state coordinates, we introduce a reversible (if possible) by $\overline{\mathbf{x}}$ function $\mathbf{r}_{\mathbf{x}}(\overline{\mathbf{x}}, \mathbf{w}, \boldsymbol{\gamma})$:

$$\mathbf{x} = \mathbf{r}_{\mathbf{x}}(\overline{\mathbf{x}}, \mathbf{w}, \boldsymbol{\gamma}), \ \mathbf{w}(t) = \mathbf{W}\left(U(t), \frac{d\mathbf{x}^*(t)}{dt}\right), \tag{29}$$

where $\mathbf{W}\left(\mathbf{U}(t), \frac{d\mathbf{x}^*(t)}{dt}\right)$—is a functional that satisfies the condition:

$$\forall \mathbf{U}(t) \equiv const \ \forall \frac{\partial}{\partial t}\left(\frac{d\mathbf{x}^*}{dt}\right) \equiv \mathbf{0} \Rightarrow \mathbf{w}(t) = \mathbf{W}\left(\mathbf{U}(t), \frac{d\mathbf{x}^*}{dt}\right) \equiv const \tag{30}$$

In this case, the expressions for the approximate solution of Equations (1) and (2) up to the coefficients $\gamma$ found from these Equations (1) and (2) are represented as:

$$\mathbf{x}(t) = \mathbf{r_x}\left(\overline{\mathbf{x}}_t(\overline{\mathbf{x}}_0, t_0, t), \mathbf{w}(t), \gamma\right), \quad \mathbf{w}(t) = \mathbf{W}\left(U(t), \frac{d\mathbf{x}^*(t)}{dt}\right). \tag{31}$$

Let us show that the analytical solution defined by (27)–(31) is the solution of Equations (1) and (2). Indeed, since $\overline{\mathbf{x}}(t) = \overline{\mathbf{x}}_t(\overline{\mathbf{x}}_0, t_0, t)$ is a solution to Equations (1) and (2) written for the topology matrix $\overline{\mathbf{B}}(\overline{\mathbf{x}}, \overline{\mathbf{U}})$, dissipative matrix $\overline{\mathbf{A}}(\overline{\mathbf{x}}, \overline{\mathbf{U}})$, and energy $\overline{W}(\overline{\mathbf{x}}, \overline{\mathbf{U}})$, as well as piecewise constant external flows $d\overline{\mathbf{x}}^*(t)/dt$, i.e.:

$$\frac{d\overline{\mathbf{x}}(t)}{dt} = \overline{\mathbf{B}}(\overline{\mathbf{x}}(t), \overline{\mathbf{U}}(t))\frac{\delta\Delta\overline{\mathbf{x}}(t)}{dt} + \frac{d\overline{\mathbf{x}}^*(t)}{dt}, \quad \frac{\delta\Delta\overline{\mathbf{x}}(t)}{dt} = \overline{\mathbf{A}}(\overline{\mathbf{x}}(t), \overline{\mathbf{U}}(t)) \cdot \Delta\overline{\mathbf{F}}(\overline{\mathbf{x}}(t), \overline{\mathbf{U}}(t)),$$

$$\Delta\overline{\mathbf{F}}(\overline{\mathbf{x}}, \overline{\mathbf{U}}) = \overline{\mathbf{B}}^{-T}(\overline{\mathbf{x}}, \overline{\mathbf{U}}) \cdot \overline{\mathbf{F}}(\overline{\mathbf{x}}, \overline{\mathbf{U}}), \quad \overline{\mathbf{F}}(\overline{\mathbf{x}}, \overline{\mathbf{U}}) = -\nabla_{\overline{\mathbf{x}}}\overline{W}(\overline{\mathbf{x}}, \overline{\mathbf{U}}),$$

and taking into account that:

$$\nabla_{\mathbf{x}}\overline{W}(\overline{\mathbf{x}}, \overline{\mathbf{U}}) = \nabla_{\mathbf{x}}\overline{W}(\mathbf{r_x}^{-1}(\mathbf{x}, \mathbf{w}, \gamma), \overline{\mathbf{U}}) = (\mathbf{J}^T_{\mathbf{r},\mathbf{x}}(\overline{\mathbf{x}}, \mathbf{w}, \gamma))^{-1}\nabla_{\overline{\mathbf{x}}}\overline{W}(\overline{\mathbf{x}}, \overline{\mathbf{U}})$$

where $\mathbf{J}_{\mathbf{r},\mathbf{x}}(\overline{\mathbf{x}}, \mathbf{w}, \gamma)$ is the Jacobian of matrix functions $\mathbf{r_x}(\overline{\mathbf{x}}, \mathbf{w}, \gamma)$ with respect to $\overline{\mathbf{x}}$, by virtue of (29) and (31) we have:

$$\frac{d\mathbf{x}(t)}{dt} = \mathbf{J}_{\mathbf{r},\mathbf{x}}(\overline{\mathbf{x}}(t), \mathbf{w}(t), \gamma)\overline{\mathbf{B}}(\overline{\mathbf{x}}(t), \overline{\mathbf{U}}(t))\frac{\delta\Delta\overline{\mathbf{x}}(t)}{dt} + \mathbf{J}_{\mathbf{r},\mathbf{x}}(\overline{\mathbf{x}}(t), \mathbf{w}(t), \gamma)\frac{d\overline{\mathbf{x}}^*(t)}{dt} + \mathbf{J}_{\mathbf{r},\mathbf{w}}(\overline{\mathbf{x}}(t), \mathbf{w}(t), \gamma)\frac{d\mathbf{w}(t)}{dt}$$

$$\frac{\delta\Delta\overline{\mathbf{x}}(t)}{dt} = \overline{\mathbf{A}}(\overline{\mathbf{x}}(t), \overline{\mathbf{U}}(t)) \cdot \Delta\overline{\mathbf{F}}(\overline{\mathbf{x}}(t), \overline{\mathbf{U}}(t))$$

$$\Delta\overline{\mathbf{F}}(\overline{\mathbf{x}}, \overline{\mathbf{U}}) = \overline{\mathbf{B}}^{-T}(\overline{\mathbf{x}}, \overline{\mathbf{U}}) \cdot \overline{\mathbf{F}}(\overline{\mathbf{x}}, \overline{\mathbf{U}}), \quad \overline{\mathbf{F}}(\overline{\mathbf{x}}, \overline{\mathbf{U}}) = -\mathbf{J}^T_{\mathbf{r},\mathbf{x}}(\overline{\mathbf{x}}, \mathbf{w}, \gamma)\nabla_{\mathbf{x}}\overline{W}(\mathbf{r_x}^{-1}(\mathbf{x}, \mathbf{w}, \gamma), \overline{\mathbf{U}}),$$

where $\mathbf{J}_{\mathbf{r},\mathbf{w}}(\overline{\mathbf{x}}, \mathbf{w}, \gamma)$ is the Jacobian matrix of the function $\mathbf{r_x}(\overline{\mathbf{x}}, \mathbf{w}, \gamma)$ with respect to $\mathbf{w}$; hence, taking into account that the parameters $\overline{\mathbf{U}}(t)$ are piecewise constant $\mathbf{U}(t)$, and hence $\overline{\mathbf{U}}$ is a function $\mathbf{U}$, having introduced the topology matrix $\mathbf{B}(\mathbf{x}, \mathbf{w}, \mathbf{U})$:

$$\mathbf{B}(\mathbf{x}, \mathbf{w}, \mathbf{U}) = \mathbf{J_r}(\mathbf{r_x}^{-1}(\mathbf{x}, \mathbf{w}, \gamma), \mathbf{w}, \gamma)\overline{\mathbf{B}}(\mathbf{r_x}^{-1}(\mathbf{x}, \mathbf{w}, \gamma), \mathbf{U}), \tag{32}$$

external flows $d\mathbf{x}^*(t)/dt$:

$$\frac{d\mathbf{x}^*(t)}{dt} = \mathbf{J}_{\mathbf{r},\mathbf{x}}(\overline{\mathbf{x}}(t), \mathbf{w}(t), \gamma)\frac{d\overline{\mathbf{x}}^*(t)}{dt} + \mathbf{J}_{\mathbf{r},\mathbf{w}}(\overline{\mathbf{x}}(t), \mathbf{w}(t), \gamma)\frac{d\mathbf{w}(t)}{dt}, \tag{33}$$

the free energy $W(\mathbf{x}, \mathbf{w}, \mathbf{U}) = \overline{W}(\mathbf{r_x}^{-1}(\mathbf{x}, \mathbf{w}, \gamma), \mathbf{U})$ and its partial derivatives $\mathbf{F}(\mathbf{x}, \mathbf{w}, \mathbf{U})$ by state coordinates $\mathbf{x}$:

$$\mathbf{F}(\mathbf{x}, \mathbf{w}, \mathbf{U}) = -\nabla_{\mathbf{x}}\overline{W}(\mathbf{r_x}^{-1}(\mathbf{x}, \mathbf{w}, \gamma), \mathbf{U}) = -\nabla_{\mathbf{x}}W(\mathbf{x}, \mathbf{w}, \mathbf{U}),$$

as well as the dissipation matrix $\mathbf{A}(\mathbf{x}, \mathbf{w}, \mathbf{U})$:

$$\mathbf{A}(\mathbf{x}, \mathbf{w}, \mathbf{U}) = \overline{\mathbf{A}}(\mathbf{r_x}^{-1}(\mathbf{x}, \mathbf{w}, \gamma), \mathbf{U}),$$

we have:

$$\frac{d\mathbf{x}(t)}{dt} = \mathbf{B}(\mathbf{x}(t), \mathbf{w}(t), \mathbf{U}(t)) \frac{\delta \overline{\Delta \mathbf{x}}(t)}{dt} + \frac{d\mathbf{x}^*(t)}{dt}, \ \frac{\delta \overline{\Delta \mathbf{x}}(t)}{dt} = \mathbf{A}(\mathbf{x}(t), \mathbf{w}(t), \mathbf{U}(t)) \cdot \Delta \mathbf{F}(\mathbf{x}(t), \mathbf{w}(t), \mathbf{U}(t)),$$

$$\Delta \mathbf{F}(\mathbf{x}, \mathbf{w}, \mathbf{U}) = \Delta \overline{\mathbf{F}}(\mathbf{r}_\mathbf{x}^{-1}(\mathbf{x}, \mathbf{w}, \boldsymbol{\gamma}), \mathbf{U}) = \mathbf{B}^T(\mathbf{x}, \mathbf{w}, \mathbf{U}) \cdot \mathbf{F}(\mathbf{x}, \mathbf{w}, \mathbf{U}), \ \mathbf{F}(\mathbf{x}, \mathbf{w}, \mathbf{U}) = -\nabla_\mathbf{x} W(\mathbf{x}, \mathbf{w}, \mathbf{U}).$$

This directly implies that the analytical solution given by (27)–(31) is the solution of Equations (1) and (2) of the mathematical prototyping method (because from (30) it can be seen that $\mathbf{w}$ can also be attributed to the parameters of $\mathbf{U}$). Thus it is possible to set such a positively defined (in the case of inertia—non-degenerate and non-negatively defined) dissipative matrix $\mathbf{A}(\mathbf{x}, \mathbf{w}, \mathbf{U})$ and such partial derivatives $\mathbf{F}(\mathbf{x}, \mathbf{w}, \mathbf{U})$ of the free energy $W(\mathbf{x}, \mathbf{w}, \mathbf{U})$ (satisfying the condition of the total differential), as well as the topology matrix $\mathbf{B}(\mathbf{x}, \mathbf{w}, \mathbf{U})$, that this general solution given by (27)–(31) will be the general solution of Equations (1) and (2) obtained for the mentioned quantities.

External energy flows in the system, in the general case, can either change the system balance parameters (for example, the internal energy and mass of the entire system), or not.

If conditions (30) are met:

- The replacement of the state coordinates does not affect the fact that the balance parameters of the system are changed by external flows, it follows from Equations (32) and (33);
- If the analytical solution in the old coordinate system tends to the stationary state from any initial state, then the solution in the new coordinate system also tends to the stationary state from any initial state, it follows from Equation (31);
- If the analytical solution in the old coordinate system satisfies the group condition, then in the new coordinate system the solution also satisfies the group condition [47].

This implies the correctness of the general analytical solution (27)–(31) of the equations of the mathematical prototyping method.

Let us replace the state coordinates $\overline{\mathbf{y}}$ and $\overline{\mathbf{z}}$ by $\boldsymbol{\xi}$ in (13)–(17) in the same way as the replacement of the state coordinates in Equations (1) and (2) was carried out:

$$\overset{-\circ}{\mathbf{y}}(\boldsymbol{\xi}_0, t_0, t, \boldsymbol{\gamma}) = \mathbf{r}_{\overline{\mathbf{y}}}(\boldsymbol{\xi}(\boldsymbol{\xi}_0, t_0, t), \boldsymbol{\gamma}), \ \overset{-\circ}{\mathbf{z}}(\boldsymbol{\xi}_0, t_0, t, \boldsymbol{\gamma}) = \mathbf{r}_{\overline{\mathbf{z}}}(\boldsymbol{\xi}(\boldsymbol{\xi}_0, t_0, t), \boldsymbol{\gamma}), \tag{34}$$

where the system of functions $\mathbf{r}_{\overline{\mathbf{y}}}(\boldsymbol{\xi}, \boldsymbol{\gamma})$ and $\mathbf{r}_{\overline{\mathbf{z}}}(\boldsymbol{\xi}, \boldsymbol{\gamma})$ is also resolvable with respect to $\boldsymbol{\xi}$; $\boldsymbol{\xi}(\boldsymbol{\xi}_0, t_0, t)$, being a general analytical solution of a piecewise simplified system of Equations (13)–(17), is defined similarly to (27) and (28):

$$\boldsymbol{\xi}(\boldsymbol{\xi}_0, t_0, t) \equiv \tilde{\boldsymbol{\xi}}(\tilde{\boldsymbol{\xi}}_{i-1}^{(0)}, t) \Leftrightarrow t \in [t_{i-1} - t_0, t_i - t_0], \ \tilde{\boldsymbol{\xi}}_i^{(0)} = \tilde{\boldsymbol{\xi}}(\tilde{\boldsymbol{\xi}}_{i-1}^{(0)}, t_i), \ \tilde{\boldsymbol{\xi}}_0^{(0)} = \boldsymbol{\xi}_0, \ i = 1, \infty, \tag{35}$$

where in $\tilde{\mathbf{x}}_i^{(0)}$, $i = 1, \infty$ the random component is additively included, and $\tilde{\boldsymbol{\xi}}(\tilde{\boldsymbol{\xi}}_0, t)$ satisfies the property of the group [47]:

$$\tilde{\boldsymbol{\xi}}_0 = \tilde{\boldsymbol{\xi}}(\tilde{\boldsymbol{\xi}}_0, 0), \ \tilde{\boldsymbol{\xi}}(\tilde{\boldsymbol{\xi}}(\tilde{\boldsymbol{\xi}}_0, \tau), t) = \tilde{\boldsymbol{\xi}}(\tilde{\boldsymbol{\xi}}_0, t + \tau). \tag{36}$$

Hence, similarly as described above, the analytical approximation of the general solution of the equations system (13)–(17) given in the form (34)–(36) is correct (follows from [47]).

Thus, the solution of the transformed Equations (13)–(17) of the mathematical prototyping method of energy processes consists in such a selection of constant parameters $\boldsymbol{\gamma}$, that residuals $\mathbf{e}_{\overline{\mathbf{y}}}(\boldsymbol{\gamma}, \boldsymbol{\xi}_0, t_0, t)$ and $\mathbf{e}_{\overline{\mathbf{z}}}(\boldsymbol{\gamma}, \boldsymbol{\xi}_0, t_0, t)$, having the meaning of fluctuations [51,52], determined based on (13)–(17) [51,52]:

$$\mathbf{e}_{\overline{\mathbf{y}}}(\boldsymbol{\gamma}, \boldsymbol{\xi}_0, t_0, t) = \frac{\partial \overset{-\circ}{\mathbf{y}}(\boldsymbol{\xi}_0, t_0, t, \boldsymbol{\gamma})}{\partial t} - \tilde{\mathbf{B}}_\mathbf{y}(\overset{-\circ}{\mathbf{y}}(\boldsymbol{\xi}_0, t_0, t, \boldsymbol{\gamma}), \overset{-\circ}{\mathbf{z}}(\boldsymbol{\xi}_0, t_0, t, \boldsymbol{\gamma}), \mathbf{U}(t)) \frac{\delta \Delta \mathbf{x}(t)}{dt} - \frac{d\overset{-*}{\mathbf{y}}(t)}{dt}, \tag{37}$$

$$\mathbf{e}_{\mathbf{z}}^-(\boldsymbol{\gamma}, \xi_0, t_0, t) = \frac{\partial \overline{\mathbf{z}}^{\,-\circ}(\xi_0, t_0, t, \boldsymbol{\gamma})}{\partial t} - \tilde{\mathbf{B}}_{\mathbf{z}}(\overline{\mathbf{y}}^{\,-\circ}(\xi_0, t_0, t, \boldsymbol{\gamma}), \overline{\mathbf{z}}^{\,-\circ}(\xi_0, t_0, t, \boldsymbol{\gamma}), \mathbf{U}(t)) \frac{\delta \Delta \mathbf{x}(t)}{dt} - \frac{d\overline{\mathbf{z}}^{\,-*}(t)}{dt}. \tag{38}$$

should not exceed in modulus a certain fraction of the mean maximum of fluctuations [51,52]. The quantities included in (37) and (38) are determined in accordance with the expressions:

$$\frac{d\overline{\mathbf{y}}^{\,-*}(t)}{dt} = \mathbf{C}_{\mathbf{y},\mathbf{x}}(\overline{\mathbf{y}}^{\,-\circ}(\xi_0, t_0, t, \boldsymbol{\gamma}), \overline{\mathbf{z}}^{\,-\circ}(\xi_0, t_0, t, \boldsymbol{\gamma}), \mathbf{U}(t)) \frac{d\mathbf{x}^*(t)}{dt} + \mathbf{C}_{\mathbf{y},\mathbf{U}}(\overline{\mathbf{y}}^{\,-\circ}(\xi_0, t_0, t, \boldsymbol{\gamma}), \overline{\mathbf{z}}^{\,-\circ}(\xi_0, t_0, t, \boldsymbol{\gamma}), \mathbf{U}(t)) \frac{d\mathbf{U}(t)}{dt} \tag{39}$$

$$\frac{d\overline{\mathbf{z}}^{\,-*}(t)}{dt} = \mathbf{C}_{\mathbf{z},\mathbf{x}}(\overline{\mathbf{y}}^{\,-\circ}(\xi_0, t_0, t, \boldsymbol{\gamma}), \overline{\mathbf{z}}^{\,-\circ}(\xi_0, t_0, t, \boldsymbol{\gamma}), \mathbf{U}(t)) \frac{d\mathbf{x}^*(t)}{dt} + \mathbf{C}_{\mathbf{z},\mathbf{U}}(\overline{\mathbf{y}}^{\,-\circ}(\xi_0, t_0, t, \boldsymbol{\gamma}), \overline{\mathbf{z}}^{\,-\circ}(\xi_0, t_0, t, \boldsymbol{\gamma}), \mathbf{U}(t)) \frac{d\mathbf{U}(t)}{dt} \tag{40}$$

$$\frac{\delta \Delta \mathbf{x}(t)}{dt} = \mathbf{A}(\overline{\mathbf{y}}^{\,-\circ}(\xi_0, t_0, t, \boldsymbol{\gamma}), \overline{\mathbf{z}}^{\,-\circ}(\xi_0, t_0, t, \boldsymbol{\gamma}), \mathbf{U}(t)) \cdot \Delta \mathbf{F}(\overline{\mathbf{y}}^{\,-\circ}(\xi_0, t_0, t, \boldsymbol{\gamma}), \overline{\mathbf{z}}^{\,-\circ}(\xi_0, t_0, t, \boldsymbol{\gamma}), \mathbf{U}(t)) \tag{41}$$

Because by selecting parameters, it is necessary to achieve values of residuals $\mathbf{e}_{\mathbf{y}}^-(\boldsymbol{\gamma}, \xi_0, t_0, t)$ and $\mathbf{e}_{\mathbf{z}}^-(\boldsymbol{\gamma}, \xi_0, t_0, t)$, not exceeding in modulus a certain fraction of the average maximum of fluctuations [51,52], and also to simultaneously minimize the objective function $Q$, determined by (24) for $\mathbf{y}(t)$, determined on the basis of (11) and (18), due to:

$$\mathbf{y}^\circ(\xi_0, t_0, t, \boldsymbol{\gamma}) = \begin{pmatrix} \overline{\mathbf{y}}^{\,-\circ T}(\xi_0, t_0, t, \boldsymbol{\gamma}) & \tilde{\mathbf{y}}^{\circ T}(\xi_0, t_0, t, \boldsymbol{\gamma}) \end{pmatrix}^T, \tilde{\mathbf{y}}^\circ(\xi_0, t_0, t, \boldsymbol{\gamma}) = \tilde{\mathbf{g}}_{\mathbf{y}}(\overline{\mathbf{y}}^{\,-\circ}(\xi_0, t_0, t, \boldsymbol{\gamma}), \overline{\mathbf{z}}^{\,-\circ}(\xi_0, t_0, t, \boldsymbol{\gamma}), \mathbf{U}(t)), \tag{42}$$

then it is advisable in terms of parameters $\boldsymbol{\gamma}$ and constant coefficients included in (25) and (26) to minimize the objective function $\tilde{Q}$ [10,51,52]:

$$\tilde{Q} = Q[\mathbf{y}(t)] + \frac{1}{2} \sum_{i=1}^{n} \sum_{l=1}^{N_{t,i}} (\mathbf{e}_{\mathbf{y}}^T(\boldsymbol{\gamma}_i, \xi_{0,i}, t_{0,i}, t_{l,i}) \mathbf{L}_{\mathbf{y}}^-(t_{l,i}) \mathbf{e}_{\mathbf{y}}^-(\boldsymbol{\gamma}_i, \xi_{0,i}, t_{0,i}, t_{l,i})) + \mathbf{e}_{\mathbf{z}}^T(\boldsymbol{\gamma}_i, \xi_{0,i}, t_{0,i}, t_{l,i}) \mathbf{L}_{\mathbf{z}}^-(t_{l,i}) \mathbf{e}_{\mathbf{z}}^-(\boldsymbol{\gamma}_i, \xi_{0,i}, t_{0,i}, t_{l,i}), \tag{43}$$

where $\mathbf{L}_{\mathbf{y}}^-(t)$, $\mathbf{L}_{\mathbf{z}}^-(t)$ are positively defined symmetric matrices, because in this case, the residuals $\mathbf{e}_{\mathbf{y}}^-(\boldsymbol{\gamma}, \xi_0, t_0, t)$ and $\mathbf{e}_{\mathbf{z}}^-(\boldsymbol{\gamma}, \xi_0, t_0, t)$, and also $\mathbf{y}(t) - \mathbf{y}^{(\ni)}(t)$ determine both the proximity of the exact solution of the system (13)–(17) to the approximate one, and the proximity of the calculated values of the characteristics of the measured parameters $\mathbf{y}(t)$ to the corresponding measured values $\mathbf{y}^{(\ni)}(t)$ of these parameters $\mathbf{y}(t)$ [10,51,53]. According to (24), Equation (43) will take the form:

$$\tilde{Q} = \frac{1}{2} \sum_{i=1}^{n} \sum_{j=1}^{N_{t,i}} \left(\mathbf{y}_i(t_j) - \mathbf{y}_i^{(E)}(t_j)\right)^T \tilde{\mathbf{L}}_i \left(\mathbf{y}_i(t_j) - \mathbf{y}_i^{(E)}(t_j)\right) +$$
$$+\frac{1}{2} \sum_{i=1}^{n} \sum_{l=1}^{N_{t,i}} \left(\mathbf{e}_{\mathbf{y}}^T(\boldsymbol{\gamma}_i, \xi_{0,i}, t_{0,i}, t_{l,i}) \mathbf{L}_{\mathbf{y}}^-(t_{l,i}) \mathbf{e}_{\mathbf{y}}^-(\boldsymbol{\gamma}_i, \xi_{0,i}, t_{0,i}, t_{l,i})\right) + \mathbf{e}_{\mathbf{z}}^T(\boldsymbol{\gamma}_i, \xi_{0,i}, t_{0,i}, t_{l,i}) \mathbf{L}_{\mathbf{z}}^-(t_{l,i}) \mathbf{e}_{\mathbf{z}}^-(\boldsymbol{\gamma}_i, \xi_{0,i}, t_{0,i}, t_{l,i}). \tag{44}$$

Thus, the construction of a model of physical and chemical processes in RES is carried out by minimizing the objective function $\tilde{Q}$, determined by virtue of (16), (17), (25), (26), (34)–(42), (44) in constant parameters $\xi_{0,i}, \boldsymbol{\gamma}_i, i = 1, n$ (at fixed initial times $t_{0,i}, i = 1, n$) and constant parameters included in (25) and (26) [40]. The number $i$ of modes can include both control and operational modes of operation. After optimizing the objective function, $\tilde{Q}$ we determine the controlled parameters $\mathbf{z}$ in the form:

$$\mathbf{z}^\circ(\xi_0, t_0, t, \boldsymbol{\gamma}) = \begin{pmatrix} \overline{\mathbf{z}}^{\,-\circ T}(\xi_0, t_0, t, \boldsymbol{\gamma}) & \tilde{\mathbf{z}}^{\circ T}(\xi_0, t_0, t, \boldsymbol{\gamma}) \end{pmatrix}^T, \tilde{\mathbf{z}}^\circ(\xi_0, t_0, t, \boldsymbol{\gamma}) = \tilde{\mathbf{g}}_{\mathbf{z}}\left(\overline{\mathbf{y}}^{\,-\circ}(\xi_0, t_0, t, \boldsymbol{\gamma}), \overline{\mathbf{z}}^{\,-\circ}(\xi_0, t_0, t, \boldsymbol{\gamma}), \mathbf{U}(t)\right). \tag{45}$$

The target function $\tilde{Q}$ includes complex expressions (25) and (26), so it is proposed to convert them to a piecewise simplified form (for example, piecewise constant). Similarly, to simplify the objective function $\tilde{Q}$ it is advisable to simplify the analytical expressions (34) of the general solution of the system of Equations (13)–(17) [40,45,54].

Thus, a generalized method of mathematical prototyping of energy processes is proposed to use as a unified approach to designing models of physical and chemical systems, in the application of which the following methods are used:

- Transformations of the coordinate system of the state space to simplify the expressions of the state functions of the properties of substances and processes;
- Transformations of the equations of the mathematical prototyping method with respect to the measured and controlled parameters;
- Splitting the space of the system state coordinate into regions in order to obtain piecewise simplified state functions for the properties of substances and processes;
- Reduction of the procedure for obtaining analytical solutions to the equations of the mathematical prototyping method for state coordinates to the problem of finding a global minimum.

## 3. Algorithm of Obtaining the Model from Experimental Data

### 3.1. The Sequence of Obtaining Controlled Parameters

So, the determination of the controlled parameters of the system by its measured parameters consists of the following sequence of actions:

For the experimentally measured parameters $\mathbf{y}$ of the RES, we perform the minimization of the objective function $\tilde{Q}$, determined by virtue of (16), (17), (25), (26), (34)–(42), (44), with respect to the parameters $\xi_{0,i}$, $\gamma_i$, $i = 1, n$, and the parameters included in (25) and (26).

For the optimized parameters $\xi_{0,i}$, $\gamma_i$, and the parameters included in (25) and (26), we compare the values of residuals $\mathbf{e}_{\underset{\mathbf{y}}{-}}(\gamma_i, \xi_{0,i}, t_{0,i}, t)$ and $\mathbf{e}_{\underset{\mathbf{z}}{-}}(\gamma_i, \xi_{0,i}, t_{0,i}, t)$ with the value of the average maximum of fluctuations; if they are higher than the average maximum of fluctuations we should conduct the correction of $\mathbf{r}_{\underset{\mathbf{y}}{-}}(\xi, \gamma)$ and $\mathbf{r}_{\underset{\mathbf{z}}{-}}(\xi, \gamma)$ then return to point 1; otherwise, go to the next item.

Substituting the optimal parameters $\xi_{0,i}$ and $\gamma_i$ into (42) and (45) we obtain the dynamics of the measured $\mathbf{y}(t) = \mathbf{y}^\circ(\xi_0, t_0, t, \gamma)$ and control $\mathbf{z}(t) = \mathbf{z}^\circ(\xi_0, t_0, t, \gamma)$ parameters of the system.

Equations (13)–(18), and hence the proposed method for determining the controlled parameters of the system from the measured ones, can be implemented based on model-based design [55] using a block diagram [37].

### 3.2. Implementation of the Algorithm for Obtaining Controlled Parameters

The practical implementation of the approach proposed in the article to building models of physical and chemical processes in RES (the method of mathematical prototyping of energy processes) involves the following sequence of actions:

Formation of a list of physical and chemical processes in the object of study.

Determination of a set of assumptions, including considered modes of operation and external disturbances.

Writing a complete system of equations for the dynamics of physical and chemical processes in accordance with the proposed method—the method of mathematical prototyping.

Determining the required set of experimental data based on analyzing the system state dependence of the properties of substances and processes.

Setting analytical expressions (25) and (26) for the properties of the substances and the processes up to constant coefficients.

Piecewise simplification of analytical expressions (25) and (26).

Identification of the coefficients of analytical expressions.

Checking the correctness of piecewise simplification procedures.

Validation of the obtained models according to the control experimental data.

## 4. Results

In the current chapter, two examples of the application described methods are presented. The first of them is the calculation of physical and chemical processes model parameters in a nickel–cadmium battery from a 20NKBN-25-U3 series of three cells. The second one is a mathematical model for the voltage and temperature of lithium-ion batteries of the US18650VTC6 series.

### 4.1. An Example of a Nickel–Cadmium Battery

4.1.1. Physical and Chemical Processes in a Nickel–Cadmium Accumulator

The electrochemical system of a nickel–cadmium accumulator is a positive nickel oxide electrode $NiOOH$ and a negative cadmium electrode immersed in an electrolyte solution $KOH$. The substance $KOH$ does not enter the reaction, it is only a carrier of hydroxide ions $OH^-$ [56].

The main current-generating process taking place on the positive nickel oxide electrode [56]:

$$NiOOH + H_2O + e^- \leftrightarrow Ni(OH)_2 + OH^- \tag{46}$$

Hydroxide ions $OH^-$ diffuse through the electrolyte and react on the negative cadmium electrode in accordance with the reaction [56]:

$$Cd + 2OH^- \leftrightarrow Cd(OH)_2 + 2e^- \tag{47}$$

When a nickel–cadmium accumulator is recharged, the process of oxygen evolution proceeds at the positive electrode [56]:

$$2OH^- \leftrightarrow \frac{1}{2}O_2 + H_2O + 2e^- \tag{48}$$

Oxygen $O_2$ diffuses through the porous separator to the negative electrode and is reduced on it [56]:

$$\frac{1}{2}O_2 + Cd + H_2O \leftrightarrow Cd(OH)_2 \tag{49}$$

Reaction (49) of oxygen reduction is exothermic, it leads to heating of the accumulator, which can cause its thermal runaway [56].

4.1.2. Assumptions Imposed on the Mathematical Model of Physicochemical Processes in the Nickel–Cadmium Accumulators

The modeling of physicochemical processes in the nickel–cadmium accumulators is carried out taking into account the following assumptions [57]:

- Aging processes in the accumulators are not modeled (due to the fact that they proceed much more slowly than main processes);
- Hydrogen release is not simulated (the nickel–cadmium accumulators are fairly new) [56];
- Distribution of water in the volume of the separator is even;
- The volume of the separator is divided into near-anode and near-cathode regions, and the state of each region is characterized by averaged values of the distributed quantities;
- Physical and chemical processes between each pair of electrodes are identical, therefore, the accumulators are represented with one pair of electrodes, on which the above processes occur;
- The temperature of the accumulators is uniform;
- Cross-diffusion of hydroxide ions $OH^-$ and oxygen molecules is absent;
- The oxygen above the separator is in equilibrium with the oxygen in the separator;
- The contact of the base of the positive and negative electrodes with their spattering is ideal;
- The "memory effect" [56] of a nickel–cadmium accumulators is not taken into account;
- Capacities of double layers of positive and negative electrodes are not taken into account.

Within the framework of these assumptions, the model of a nickel–cadmium battery is built using the mathematical prototyping methods of energy processes [57].

4.1.3. Mathematical Model of Physical and Chemical Processes in a Nickel–Cadmium Accumulator

The mathematical model of a nickel–cadmium accumulator includes the following quantities [57–59]:

- Charge transferred through an external circuit, $\Delta q$, A·h;
- Current in the external circuit, $I = \delta\Delta q/dt$, A;
- Current through the membrane $\delta\Delta q_m/dt$, A
- Component of the current in the external circuit, due to the main current-generating processes, $\delta(\Delta q)_{base}/dt$, A;
- Charge accumulated in the membrane $q_{acc}$, A·h;
- Component of the current in the external circuit, due to the release of oxygen, $\delta(\Delta q)_{O_2}/dt$, A;
- Membrane resistance $R_m$, Ohm;
- Membrane capacity $C_m$, F;
- Internal resistance due to the main current-generating processes $r_{in.base}$, Ohm;
- Internal resistance due to the release of oxygen $r_{in.O_2}$, Ohm;
- EMF of the main current-generating processes $E_{base}$, V;
- EMF due to the release of oxygen $E_{O_2}$, V;
- Cross EMF for the main current, due to the release of oxygen $E_{O_2}^{base}$, V;
- The coefficient of cross-EMF for the main current, due to the release of oxygen $\varepsilon_{O_2}^{base}$, V;
- Cross EMF for the current associated with the release of oxygen, due to the main current, $E_{base}^{O_2}$, V;
- Cross-EMF coefficient for the current associated with the release of oxygen, due to the main current, $\varepsilon_{base}^{O_2}$, V;
- Electrical potentials of the positive $\phi^+$ and negative $\phi^-$ electrodes, respectively, V;
- The number of moles of accumulated oxygen in the electrode regions of the positive $\Delta\nu_{O_2}^+$ and negative $\Delta\nu_{O_2}^-$ electrodes, respectively;
- The number of moles of oxygen released at the positive electrode $\delta\left(\Delta\nu_{O_2}^+\right)_c$ and utilized at the negative electrode $\delta\left(\Delta\nu_{O_2}^-\right)_c$;
- The number of moles of oxygen diffused through the membrane $\delta\Delta\nu_{O_2}$;
- Chemical potentials of oxygen in the anode region $\mu_{O_2}^+$ and cathode region $\mu_{O_2}^-$, J/mol;
- Equilibrium chemical potential of oxygen $\mu_{O_2}^*$, J/mol;
- Thermal coefficient for the main current $\widetilde{q}_{base}$;
- Thermal coefficient for the main current associated with the release of oxygen at the positive electrode $\widetilde{q}_{O_2+}$;
- Thermal coefficients of oxygen diffusion through the electrolyte membrane $\widetilde{q}_{O_2d}$ and oxygen utilization at the negative electrode $\widetilde{q}_{O_2u}$;
- Heat capacity of the accumulator, $C_p$, J/K;
- Heat release power in the accumulator, $Q_V$, W;
- Heat transfer coefficient of the accumulator, $K$, W/(m²·K);
- Heat transfer area, $S$, m²;
- Accumulator temperature, $T$, ambient temperature, $T_0$, K.

The mentioned mathematical model of the physical and chemical processes dynamics in a nickel–cadmium accumulator, obtained by mathematical prototyping, has the following form [57–59]:

- Stoichiometric ratios:

$$\delta\Delta\nu_{Ni(OH)_2} = F\delta(\Delta q)_{base}, \ \delta\left(\Delta\nu_{O_2}^+\right)_c = -\tfrac{1}{4F}\delta(\Delta q)_{O_2},$$

$$\delta\Delta\nu_{Cd(OH)_2} = \tfrac{1}{2}F\left(\delta(\Delta q)_{base} + \delta(\Delta q)_{O_2}\right) + 2\delta\left(\Delta\nu_{O_2}^-\right)_c,$$

- Equations of the conservation law of oxygen mass:

$$\delta\Delta\nu_{O_2}^+ = \delta\left(\Delta\nu_{O_2}^+\right)_c - \delta\Delta\nu_{O_2}, \ \delta\Delta\nu_{O_2}^- = \delta\Delta\nu_{O_2} - \delta\left(\Delta\nu_{O_2}^-\right)_c;$$

- Equivalent circuit equations for electrochemical processes (Figure 1):

$$r_{in.base}\frac{\delta(\Delta q)_{base}}{dt} + R_m\frac{\delta \Delta q_m}{dt} + (\phi^+ - \phi^-) = E_{base} - E_{O_2}^{base},$$

$$r_{base.O_2}\frac{\delta(\Delta q)_{O_2}}{dt} + R_m\frac{\delta \Delta q_m}{dt} + (\phi^+ - \phi^-) = E_{O_2} - E_{base}^{O_2}, \quad E_{O_2} = \frac{\mu_{O_2}^+ - \mu_{O_2}^*}{4F},$$

$$E_{O_2}^{base} = \varepsilon_{O_2}^{base}\frac{\delta(\Delta q)_{O_2}}{dt}, \quad E_{base}^{O_2} = \varepsilon_{base}^{O_2}\frac{\delta(\Delta q)_{base}}{dt}, \quad \varepsilon_{O_2}^{base} = \varepsilon_{base}^{O_2},$$

$$I = \frac{\delta \Delta q}{dt} = \frac{\delta \Delta q_m}{dt} + \frac{\delta \Delta q_{acc}}{dt} = \frac{\delta(\Delta q)_{base}}{dt} + \frac{\delta(\Delta q)_{O_2}}{dt}, \quad R_m\frac{\delta \Delta q_m}{dt} = \frac{q_{acc}}{C_m};$$

- Oxygen diffusion equations and oxygen utilization equations:

$$\frac{\delta \Delta v_{O_2}}{dt} = D_{O_2}\left(\mu_{O_2}^+ - \mu_{O_2}^-\right), \quad \frac{d\left(\Delta v_{O_2}^-\right)_c}{dt} = R_{O_2}\left(\mu_{O_2}^- - \mu_{O_2}^*\right);$$

- Heat release power:

$$Q_V = \widetilde{q}_{base}r_{in.base}\left(\frac{d(\Delta q)_{base}}{dt}\right)^2 + \widetilde{q}_{O_2+}r_{in.O_2}\left(\frac{d(\Delta q)_{O_2}}{dt}\right)^2 + R_m\left(\frac{d\Delta q_m}{dt}\right)^2 + \widetilde{q}_{O_2d}\frac{1}{D_{O_2}}\left(\frac{d\Delta v_{O_2}}{dt}\right)^2 +$$

$$+ \left(\widetilde{q}_{base}\varepsilon_{O_2}^{base} + \widetilde{q}_{O_2+}\varepsilon_{base}^{O_2}\right)\frac{d(\Delta q)_{O_2}}{dt}\frac{d(\Delta q)_{base}}{dt} + \widetilde{q}_{O_2u}R_{O_2}\left(\frac{d\left(\Delta v_{O_2}^-\right)_c}{dt}\right)^2;$$

- Equations of thermal coefficients:

$$\widetilde{q}_{base} = 1 - \frac{T\frac{\partial E_{base}}{\partial T}}{E_{base} - \frac{q_{acc}}{C_m} - (\phi^+ - \phi^-)}, \quad \widetilde{q}_{O_2+} = 1 - \frac{T\frac{\partial E_{O_2}}{\partial T}}{E_{O_2} - \frac{q_{acc}}{C_m} - (\phi^+ - \phi^-)},$$

$$\widetilde{q}_{O_2d} = 1 - \frac{T\frac{\partial\left(\mu_{O_2}^+ - \mu_{O_2}^-\right)}{\partial T}}{\left(\mu_{O_2}^+ - \mu_{O_2}^-\right)}, \quad \widetilde{q}_{O_2u} = 1 - \frac{T\frac{\partial\left(\mu_{O_2}^- - \mu_{O_2}^*\right)}{\partial T}}{\left(\mu_{O_2}^- - \mu_{O_2}^*\right)};$$

- Heat balance equation:

$$Q_V = C_p\frac{dT}{dt} + KS(T - T_0)$$

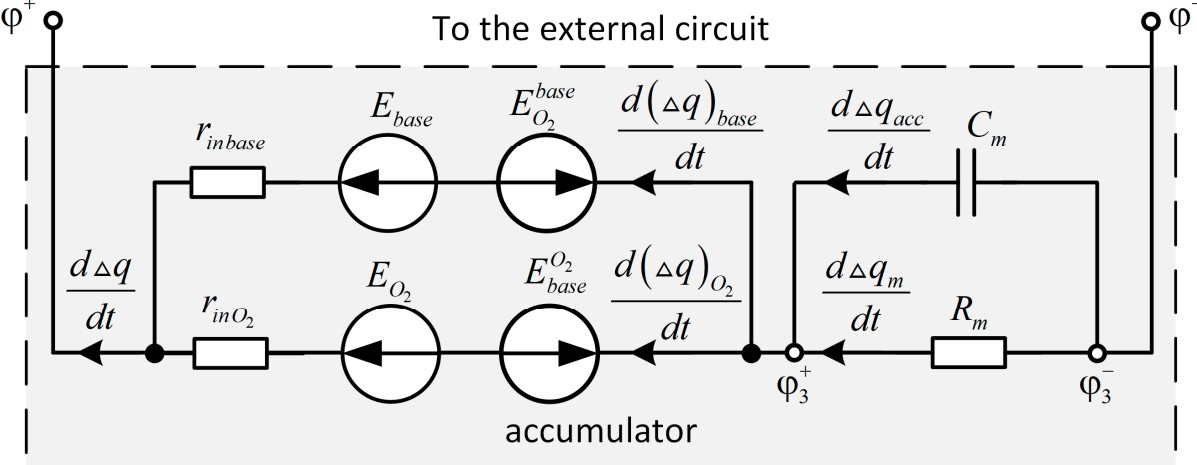

**Figure 1.** Equivalent circuit for electrochemical processes in a nickel–cadmium accumulator.

The resulting system of equations is closed, and we will be able to predict the physical and chemical processes in a nickel–cadmium accumulator [57–59] by knowing the following parameters:

- The parameters of the equivalent circuit (Figure 1);
- The function of the oxygen chemical potential;
- The oxygen chemical potential at equilibrium;
- Coefficients of diffusion and utilization of oxygen.

### 4.1.4. Identification of the Substances and Processes Properties in a Nickel–Cadmium Battery

The identification of the normalized variables is completed based on the following assumptions (and simplifications) [57,59]:

- The oxygen cycle is assumed to be stationary—how much oxygen was released on the nickel oxide electrode, the same amount diffused to the cadmium electrode, the same amount was utilized on the electrode;
- The diffusion coefficient of oxygen through the membrane is constant;
- The main current-forming reactions (46) and (47) in the forward direction proceed on the regions of the corresponding electrodes free from the hydroxide film, in the reverse direction, on the regions covered with the hydroxide film (Figure 2);
- The reaction of release (48) and utilization (49) of oxygen on the corresponding electrodes proceeds only on the areas of these electrodes free from the hydroxide film (Figure 2);
- The area of the hydroxide film covering the electrodes is directly proportional to the number of moles of the corresponding hydroxides (Figure 2);
- The reactivity coefficients of the main current-forming reactions do not depend on the number of moles of oxygen in the near-electrode region;
- The capacity and resistance of the membrane do not depend on the current $\delta \Delta q_m / dt$ and the redistributed charge $q_{acc}$;
- The dependences of the reactivity coefficients of electrochemical reactions on the redistribution of the electrolyte in the active sites are similar to each other;
- The parameters of a nickel–cadmium accumulator at temperatures below the critical temperature at which the development of thermal acceleration begins do not depend on temperature;
- The heat capacity and heat transfer coefficient of the battery are constant throughout the charge/discharge process.

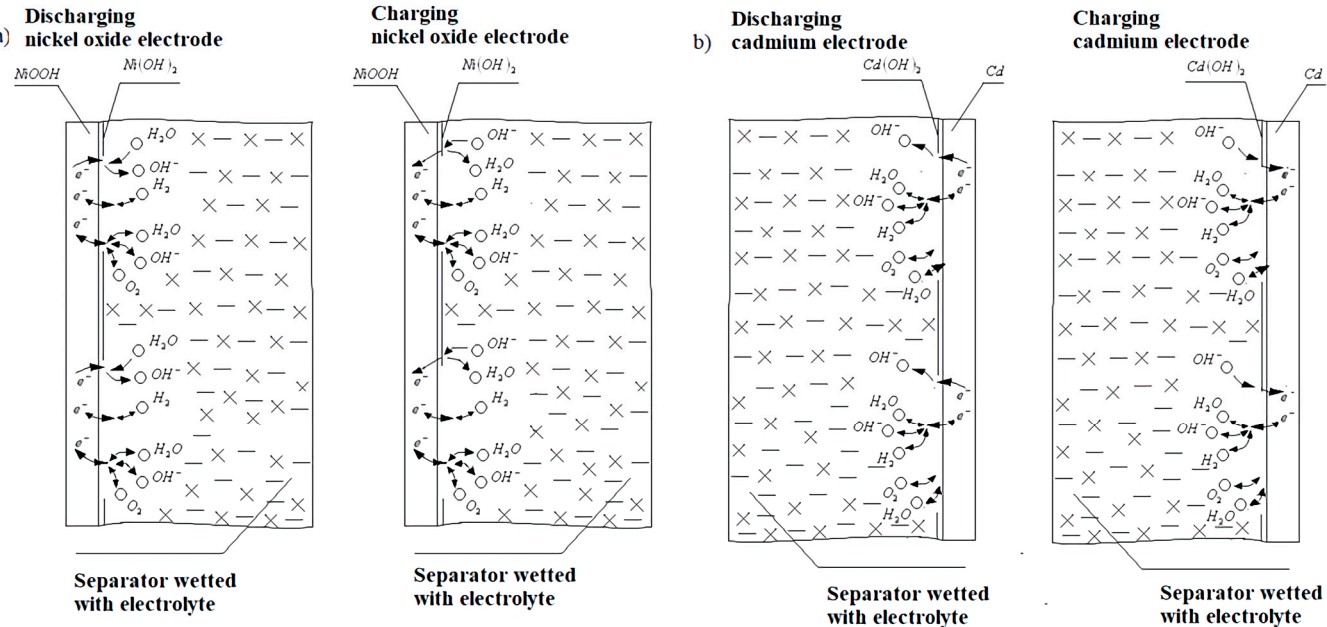

**Figure 2.** Active sites of nickel oxide (**a**) and cadmium (**b**) electrodes.

The assumptions formulated above related to the coating of the electrodes with a hydroxide film are taken into account when obtaining analytical dependences of the resistances $r_{in.base}$, $r_{in.O_2}$ and the coefficient of transfer EMF $\varepsilon_{base}^{O_2} = \varepsilon_{O_2}^{base}$ on the battery discharge level [57]. The coefficients of the transfer EMF do not depend on the degree of coating of the electrodes, and the resistances are $r_{in.base}$, $r_{in.O_2}$ inversely proportional to the

active areas of the electrodes, which means they are inversely proportional to the numbers of moles of hydroxide films covering the electrodes [57]. Hence, the analytical functions of the properties of the main current-forming reactions (46) and (47), as well as the oxygen cycle reactions (48) and (49), it is advisable to set in the form (depending on the charge or discharge of a nickel–cadmium battery) [57]:

$$
r_{in.base} = \begin{cases}
\dfrac{r^{\circ d}_{in.base}+\Delta r^{\circ d}_{in.base}}{2\left(1-\dfrac{C_0^++\Delta q+4F\left(\Delta \nu_{O_2}^+\right)_c}{C^+}\right)} + \dfrac{r^{\circ d}_{in.base}-\Delta r^{\circ d}_{in.base}}{2\left(1-\dfrac{C_0^-+\Delta q+4F\left(\Delta \nu_{O_2}^-\right)_c}{C^-}\right)}, & \begin{cases} \dfrac{d(\Delta q)_{base}}{dt} \geq 0 \\ \dfrac{d(\Delta q)_{base}}{dt}+\dfrac{d(\Delta q)_{O_2}}{dt} \geq 0 \end{cases} \\[2em]
\dfrac{r^{\circ d}_{in.base}+\Delta r^{\circ d}_{in.base}}{2\left(1-\dfrac{C_0^++\Delta q+4F\left(\Delta \nu_{O_2}^+\right)_c}{C^+}\right)} + \dfrac{r^{\circ c}_{in.base}-\Delta r^{\circ c}_{in.base}}{2\dfrac{C_0^-+\Delta q+4F\left(\Delta \nu_{O_2}^-\right)_c}{C^-}}, & \begin{cases} \dfrac{d(\Delta q)_{base}}{dt} \geq 0 \\ \dfrac{d(\Delta q)_{base}}{dt}+\dfrac{d(\Delta q)_{O_2}}{dt} < 0 \end{cases} \\[2em]
\dfrac{r^{\circ c}_{in.base}+\Delta r^{\circ c}_{in.base}}{2\dfrac{C_0^++\Delta q+4F\left(\Delta \nu_{O_2}^+\right)_c}{C^+}} + \dfrac{r^{\circ d}_{in.base}-\Delta r^{\circ d}_{in.base}}{2\left(1-\dfrac{C_0^-+\Delta q+4F\left(\Delta \nu_{O_2}^-\right)_c}{C^-}\right)}, & \begin{cases} \dfrac{d(\Delta q)_{base}}{dt} < 0 \\ \dfrac{d(\Delta q)_{base}}{dt}+\dfrac{d(\Delta q)_{O_2}}{dt} \geq 0 \end{cases} \\[2em]
\dfrac{r^{\circ c}_{in.base}+\Delta r^{\circ c}_{in.base}}{2\dfrac{C_0^++\Delta q+4F\left(\Delta \nu_{O_2}^+\right)_c}{C^+}} + \dfrac{r^{\circ c}_{in.base}-\Delta r^{\circ c}_{in.base}}{2\dfrac{C_0^-+\Delta q+4F\left(\Delta \nu_{O_2}^-\right)_c}{C^-}}, & \begin{cases} \dfrac{d(\Delta q)_{base}}{dt} < 0 \\ \dfrac{d(\Delta q)_{base}}{dt}+\dfrac{d(\Delta q)_{O_2}}{dt} < 0 \end{cases}
\end{cases}
\tag{50}
$$

$$
r_{in.O_2} = \begin{cases}
\dfrac{2r^{\circ d}_{in.O_2}-r^{\circ d}_{in.base}+\Delta r^{\circ d}_{in.base}}{2\left(1-\dfrac{C_0^++\Delta q+4F\left(\Delta \nu_{O_2}^+\right)_c}{C^+}\right)} + \dfrac{r^{\circ d}_{in.base}-\Delta r^{\circ d}_{in.base}}{2\left(1-\dfrac{C_0^-+\Delta q+4F\left(\Delta \nu_{O_2}^-\right)_c}{C^-}\right)}, & \begin{cases} \dfrac{d(\Delta q)_{O_2}}{dt} \geq 0 \\ \dfrac{d(\Delta q)_{base}}{dt}+\dfrac{d(\Delta q)_{O_2}}{dt} \geq 0 \end{cases} \\[2em]
\dfrac{2r^{\circ d}_{in.O_2}-r^{\circ d}_{in.base}+\Delta r^{\circ d}_{in.base}}{2\left(1-\dfrac{C_0^++\Delta q+4F\left(\Delta \nu_{O_2}^+\right)_c}{C^+}\right)} + \dfrac{r^{\circ c}_{in.base}-\Delta r^{\circ c}_{in.base}}{2\dfrac{C_0^-+\Delta q+4F\left(\Delta \nu_{O_2}^-\right)_c}{C^-}}, & \begin{cases} \dfrac{d(\Delta q)_{O_2}}{dt} \geq 0 \\ \dfrac{d(\Delta q)_{base}}{dt}+\dfrac{d(\Delta q)_{O_2}}{dt} < 0 \end{cases} \\[2em]
\dfrac{2r^{\circ c}_{in.O_2}-r^{\circ c}_{in.base}+\Delta r^{\circ c}_{in.base}}{2\left(1-\dfrac{C_0^++\Delta q+4F\left(\Delta \nu_{O_2}^+\right)_c}{C^+}\right)} + \dfrac{r^{\circ d}_{in.base}-\Delta r^{\circ d}_{in.base}}{2\left(1-\dfrac{C_0^-+\Delta q+4F\left(\Delta \nu_{O_2}^-\right)_c}{C^-}\right)}, & \begin{cases} \dfrac{d(\Delta q)_{O_2}}{dt} < 0 \\ \dfrac{d(\Delta q)_{base}}{dt}+\dfrac{d(\Delta q)_{O_2}}{dt} \geq 0 \end{cases} \\[2em]
\dfrac{2r^{\circ c}_{in.O_2}-r^{\circ c}_{in.base}+\Delta r^{\circ c}_{in.base}}{2\left(1-\dfrac{C_0^++\Delta q+4F\left(\Delta \nu_{O_2}^+\right)_c}{C^+}\right)} + \dfrac{r^{\circ c}_{in.base}-\Delta r^{\circ c}_{in.base}}{2\dfrac{C_0^-+\Delta q+4F\left(\Delta \nu_{O_2}^-\right)_c}{C^-}}, & \begin{cases} \dfrac{d(\Delta q)_{O_2}}{dt} < 0 \\ \dfrac{d(\Delta q)_{base}}{dt}+\dfrac{d(\Delta q)_{O_2}}{dt} < 0 \end{cases}
\end{cases}
\tag{51}
$$

$$
\varepsilon_{O_2}^{base} = \varepsilon_{base}^{O_2} = \begin{cases}
\dfrac{r^{\circ d}_{in.base}-\Delta r^{\circ d}_{in.base}}{2\left(1-\dfrac{C_0^-+\Delta q+4F\left(\Delta \nu_{O_2}^-\right)_c}{C^-}\right)}, & \dfrac{d(\Delta q)_{base}}{dt}+\dfrac{d(\Delta q)_{O_2}}{dt} \geq 0 \\[2em]
\dfrac{r^{\circ c}_{in.base}-\Delta r^{\circ c}_{in.base}}{2\dfrac{C_0^-+\Delta q+4F\left(\Delta \nu_{O_2}^-\right)_c}{C^-}}, & \dfrac{d(\Delta q)_{base}}{dt}+\dfrac{d(\Delta q)_{O_2}}{dt} < 0
\end{cases}
\tag{52}
$$

$$
R_{O_2} = R_{O_2}^{\circ}\left(1-\dfrac{C_0^-+\Delta q+4F\left(\Delta \nu_{O_2}^-\right)_c}{C^-}\right)
\tag{53}
$$

The parameters included in the above dependencies are taken only for pure nickel oxide and cadmium electrodes [57].

When identifying the parameters included in (50)–(53), the discharge curve (the space of state coordinates) of a nickel–cadmium accumulator is represented as three sections (Figure 3) [57]:

- Section I (Figure 3b) corresponds to a decrease in the discharge voltage associated with the polarization (redistribution) of the electrolyte;
- In sections II and III (Figure 3b), the battery is discharged solely by filling the electrodes with products of the corresponding electrochemical reactions (hydroxide films).
- Section II differs from section III of the discharge curve in Figure 3b. In section II, an electrode with a smaller capacity is slightly coated with a hydroxide film, while in section III, the electrode is already significantly coated with a hydroxide film [57]. In section III, each fraction of the remaining small free area is already more significant than in section II, therefore, the voltage decreases much faster, and the rate of decrease increase [57].

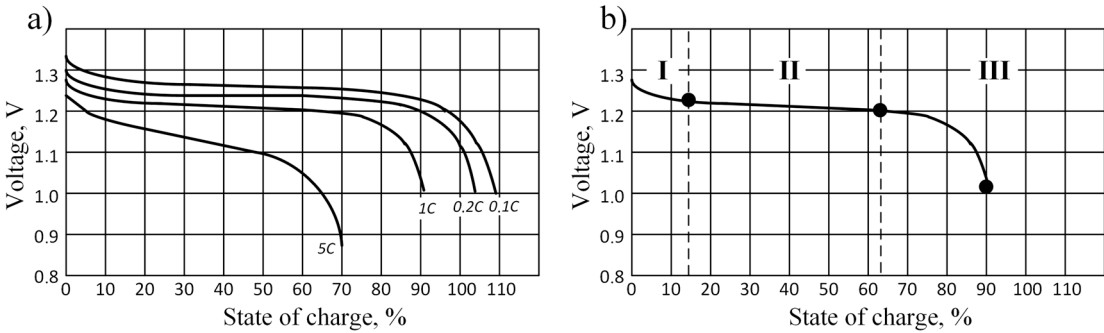

**Figure 3.** Characteristic view of discharge curves of nickel–cadmium batteries (**a**) and characteristic sections of the discharge curve of a nickel–cadmium battery (**b**).

Identification of the properties of substances and processes in a nickel–cadmium battery is carried out from the equations of physical and chemical processes by performing the following actions [57,59]:

1. The resistances of the free sections of the electrodes and the EMF of the electrode reactions are determined from the condition of coincidence of the calculated voltage curve with the experimental one for the considered discharge current and ambient temperature in sections II and III (Figure 3) corresponding to the steady-state concentrations of the electrolyte.
2. The capacity of the membrane is determined from the condition of coincidence of the calculated voltage curve with the experimental one for the discharge current and ambient temperature in section I (Figure 3).
3. The distribution of electrolyte concentrations over near-electrode regions, currents through the membrane during polarization, and the heat release power in a nickel–cadmium battery are determined for the specified discharge current of a nickel–cadmium battery and ambient temperature.
4. The heat capacity and heat transfer coefficient of a nickel–cadmium battery are determined from the condition of coincidence of the calculated temperature dynamics with the experimental one.
5. The following dependences are constructed:

    a. The membrane capacity on the current through the membrane and the temperature of the battery;
    b. The resistances of the active sections of the electrode on the charge or discharge currents;
    c. The EMF of the double layers on the redistribution of electrolyte concentrations.
6. Analytical expressions of the properties of substances and processes in a nickel–cadmium battery are constructed by adding additional components in (50)–(53).

7. The additional coefficients are determined from the discharge voltage and temperature dynamics for different discharge modes of a nickel–cadmium battery.

Thus, the identification of the parameters of the nickel–cadmium battery model is carried out using a piecewise analytical solution of the equations of dynamics of physical and chemical processes in the battery [57].

As an example, let us consider the calculation of physical and chemical processes model parameters in a nickel–cadmium battery from a 20NKBN-25-U3 series 3 battery. The discharge curves (Figure 4) show the calculated curves for the identified values for sections II and III of internal resistances and EMF without taking into account polarization. In addition, the graphs (Figure 5) show the calculated curves, which additionally take into account the membrane capacity identified in section I. The coincidence of the calculated and experimental data, as can be seen in Figure 5, confirms the sufficiently high accuracy of the calculation of the proposed method of mathematical prototyping [57].

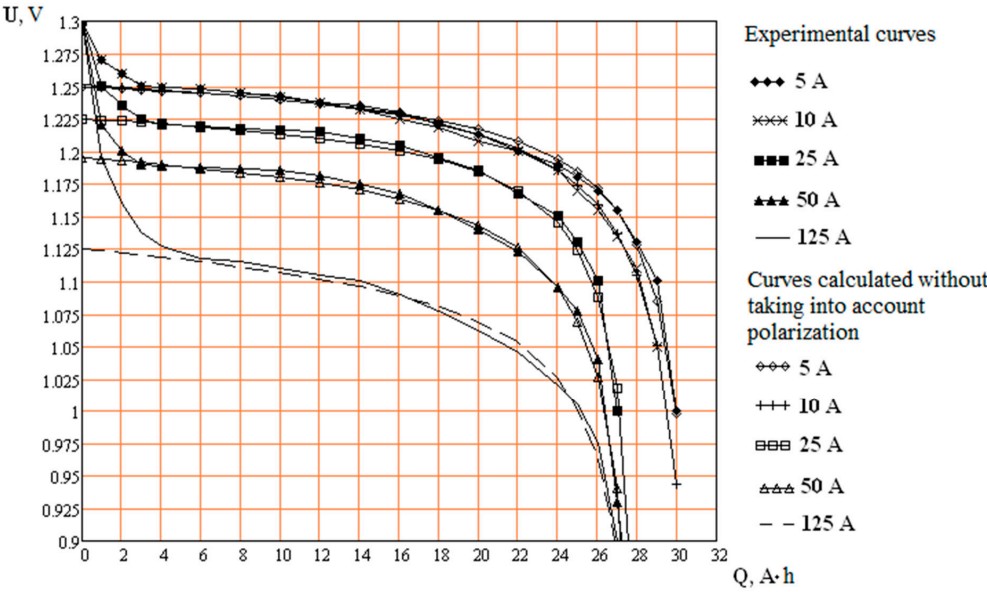

**Figure 4.** Calculation of internal resistances and EMF of a nickel–cadmium battery by comparing the calculated data with the experiment.

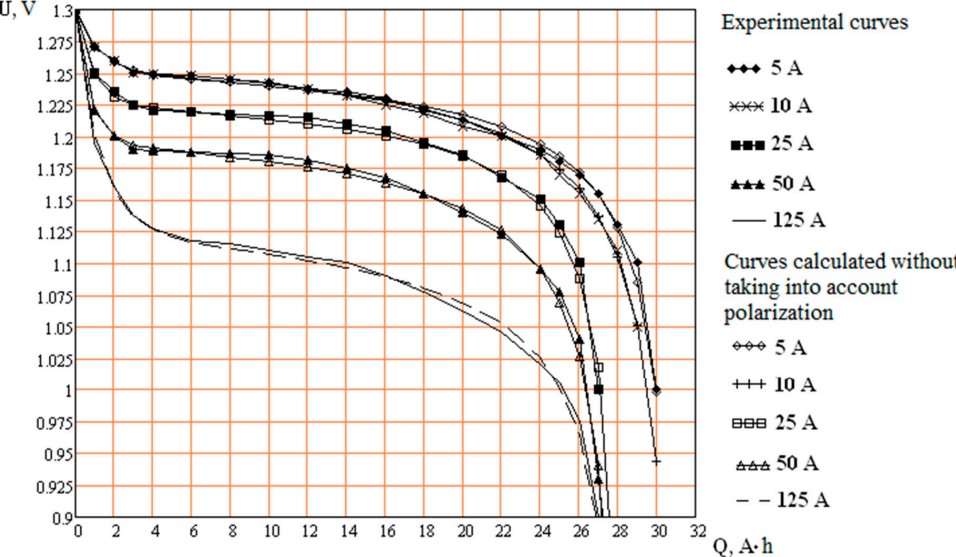

**Figure 5.** Calculation of the membrane capacities of a nickel–cadmium battery by comparing the calculated data with the experiment.

For different discharge currents, the dependence of the resistances of pure electrodes is shown in Figure 6 [57]. As it is easy to see from Figure 6, such resistances fall with increasing discharge current, which corresponds to the theoretical provisions of electrochemistry. Additionally, with an increase in the discharge current, the capacity of the membrane decreases (Figure 7) [57]. Such dependencies can be considered as dependencies on the corresponding currents since these currents in the steady state are equal to discharge currents.

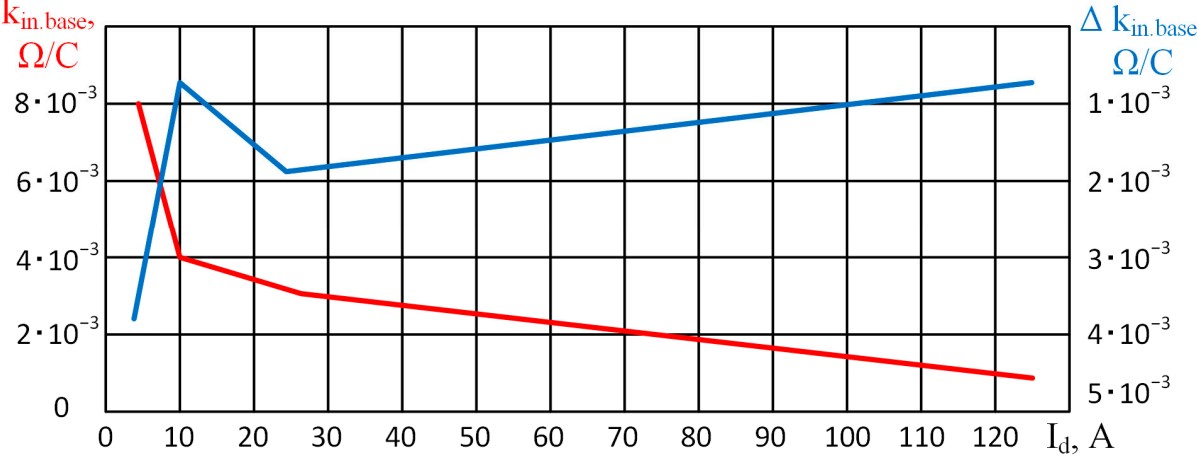

**Figure 6.** Dependence of internal resistance parameters on discharge current.

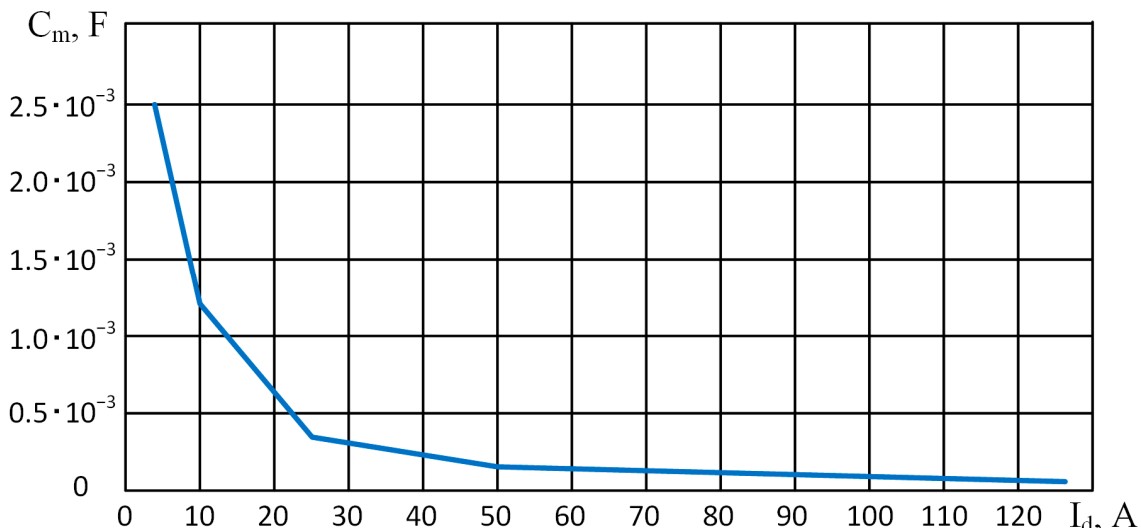

**Figure 7.** Dependence of the membrane capacity on the discharge current.

Taking into account the dependencies shown in Figures 6 and 7 in (50)–(53), as well as adding and identifying additional components in (50)–(53) (similarly to (25) and (26)), we will obtain a more complete model of physicochemical processes in a nickel–cadmium battery.

Such a model makes it possible, for example, to predict the temperature of a nickel–cadmium battery, thereby predicting and taking measures to prevent the thermal runaway of the battery [57,59].

### 4.2. Lithium-Ion Accumulator Simulation

The principle of operation of lithium-ion batteries is the intercalation/deintercalation of lithium ions into the electrodes (Figure 8) [22,60]. Similarly, to a nickel–cadmium battery, as the cells in the electrode are filled with lithium ions, the active area of the electrodes decreases. Based on this, in [61], a mathematical model for the voltage of a lithium-ion battery of the QL079KM series was obtained from the discharge voltage curves.

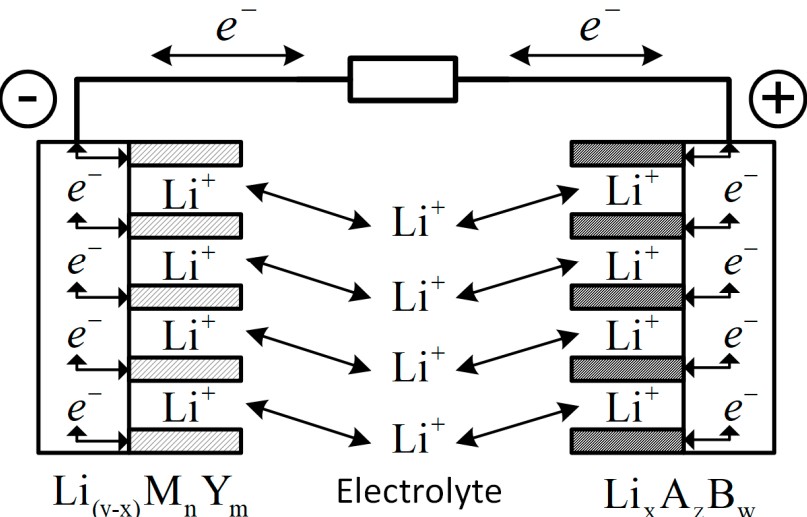

**Figure 8.** The principle of operation of a lithium-ion battery.

A mathematical model for the voltage and temperature of lithium-ion batteries of the US18650VTC6 series was obtained in [61,62] from test results similar to a nickel–cadmium battery. The largest relative error of this model did not exceed 12% [62]. Analytical expressions (25) and (26) in [62] were given by further modification of the Balter–Vollmer model [22,63].

## 5. Discussion

In this paper, a unified approach to designing models of physical and chemical processes is proposed—a generalized method of mathematical prototyping of energy processes.

The model obtained by mathematical prototyping incorporates the laws of thermodynamics, conservation laws, as well as some physical features of the processes in a RES. This guarantees the correctness of the desired model, i.e., its consistency with its general physical laws.

Obtaining a model of an arbitrary system is reduced to specifying classes of analytical expressions of properties of substances and processes satisfying the corresponding constraints with accuracy up to experimentally determined constant coefficients. The specified classes of analytical expressions cover the entire space of functions of the properties of substances and processes of the system, taking into account restrictions. The determination of the controlled system's parameters by the measured parameters is reduced to the identification of constant coefficients of the specified analytical expressions from experimental data.

To simplify the integration of equations of the mathematical prototyping method in order to reduce computational costs, an analytical task for solving differential equations of the mathematical prototyping method with an accuracy of constant coefficients is proposed.

The proposed approach to piecewise simplification of analytical expressions also reduces computational costs. Additionally, the possibility of parallelization of calculations that appears as a result of piecewise analytical simplification significantly speeds up the execution of calculations.

The correctness of the models obtained by the proposed method of mathematical prototyping for specific instances of objects allows using this approach for:

- Formation and refinement of real-time digital twins of objects and systems;
- Synthesis of objects and systems governing laws;
- Diagnostics and forecasting of the technical condition of systems, as well as medical diagnostics;
- To form and optimize technological processes (in operation and maintenance, biochemistry and bioengineering, geoengineering and meteorology, aerospace technologies, etc.);

- Designing new facilities and systems.

The physicality of the proposed method of mathematical prototyping allows using artificial intelligence methods to obtain models that do not contradict physics, unlike classical methods of constructing simulation models. The proposed piecewise analytical approach allows the processing of experimental data, and hence the training of models, in parts.

**Author Contributions:** Conceptualization, S.K.; Methodology, S.K. and I.S.; Software, I.S.; Formal analysis, I.A.; Investigation, I.S.; Data curation, I.A.; Writing—original draft, I.S.; Writing—review & editing, S.K. and I.A.; Supervision, S.K. All authors have read and agreed to the published version of the manuscript.

**Funding:** This research received no external funding.

**Conflicts of Interest:** The authors declare no conflict of interest.

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
