# Peer review of "Generalized Method of Mathematical Prototyping of Energy Processes for Digital Twins Development"

_energies, doi:10.3390/en16041933_

Round 1

Reviewer 1 Report

This paper fully illustrates the importance of reliable mathematical models for the construction of Digital Twin systems, and builds mathematical models based on the basic laws followed by the physical and chemical processes of the system, such as the law of conservation, internal perturbations, and the connection between the rates of physical and chemical processes in the system. The topic of this paper is novel, and the connection between digital twins with electrochemical systems can be better used as the basis for intelligent energy systems. The article provides a detailed derivation of the formulas in the mathematical model, and the topic selection is well answered. However, there are some shortcomings that could be further corrected. As follows:

1.The prepositions in article titles should be lowercase.

2.In this paper,the citations such as: [4,8–10,22,23], [10,24–29,33,35–37] are not prescriptive. It is best to have only one reference after each sentence.

3. The content of this paper is not closely related to digital twins, there are no diagrams and concepts related to digital twins, and only digital twins are mentioned in the abstract and discussion parts.

4. There are too many formulas from references, such as (5)-(22) is the cited document [38], (23)-(24) cited document [10, 11, 37], (25)-(26) cited document [37], etc. more of your own work should be added to the paper.

5. Three methods for finding the analytical expression of a mathematical model are given, but the process of selection is not written.

6. The introduction to equation (26) is about the topological matrix B(y(—),z(—),U), but the equation appears H(y(—),z(—),U).

7. The comma format of reference [36,37] in the 241-line is different from the others.

8. The paper format should be further revised.

Author Response

Above all, we would like to thank you for reviewing our article and we appreciate your reasonable suggestions. All your suggestions have been carefully considered and worked out. The answers and explanation of our point of view are presented below:

1.The prepositions in article titles should be lowercase.

It is fully fixed in the article.

2.In this paper,the citations such as: [4,8–10,22,23], [10,24–29,33,35–37] are not prescriptive. It is best to have only one reference after each sentence.

We have written multiple citations to provide the readers with the experience of a wide range of researchers, to present published books and articles that support our claims through a variety of approaches.

  1. The content of this paper is not closely related to digital twins, there are no diagrams and concepts related to digital twins, and only digital twins are mentioned in the abstract and discussion parts.

The article is not devoted to digital twins designing as a system that includes a mathematical model, a computer model, and information channels for communication with an object. Instead, the article aims to obtain a generalized approach to creating a mathematical basis for digital twins.

The following publications are planned to present the application of the described approach in digital twins designing at the stages of the life cycle of specific products.

  1. There are too many formulas from references, such as (5)-(22) is the cited document [38], (23)-(24) cited document [10, 11, 37], (25)-(26) cited document [37], etc. more of your own work should be added to the paper.

Due to the fact that this article is based on many years of scientific research, the results of which have been published in the form of a number of monographs and extended articles, it is impossible to fit all these materials into one article. Therefore, the article provides citations to these works.

  1. Three methods for finding the analytical expression of a mathematical model are given, but the process of selection is not written.

The choice of a specific method depends on the problem being solved, including the specific properties of the object (the level of non-linearity, non-stationarity, anisotropy, etc.). The researcher makes this choice.

  1. The introduction to equation (26) is about the topological matrix B(y(—),z(—),U), but the equation appears H(y(—),z(—),U).

The matrix H represents only the independent components of the matrix B, hence we can write that

dim(H)≤dim(B)+dim(z)

  1. The comma format of reference [36,37] in the 241-line is different from the others.

It is fully fixed in the article.

  1. The paper format should be further revised.

The structure of the article (section titles) has been refined

Reviewer 2 Report

In present paper the authors present a method of energy processes mathematical prototyping – an overall approach to modeling processes of various physical and chemical natures based on modern non-equilibrium thermodynamics, mechanics, and electrodynamics. This is a very meaningful work.

The following aspects need to be improved in the paper:

1. The accuracy of the digital twin system depends on various constant coefficients. How can the accuracy of the coefficients be guaranteed? Is there a system that does not rely on precision coefficients? such as artificial intelligence methods?

2.Language can be further improved.

Author Response

Above all, we would like to thank you for reviewing our article and we appreciate your reasonable suggestions. All your suggestions have been carefully considered and worked out. The answers and explanation of our point of view are presented below:

  1. The accuracy of the digital twin system depends on various constant coefficients. How can the accuracy of the coefficients be guaranteed? Is there a system that does not rely on precision coefficients? such as artificial intelligence methods?

The article proposes to reduce the problem of identification of nonlinear functions to the problem of parametric identification (determination of coefficients), which significantly simplifies and reduces the time to obtain a digital twin model. During exploitation, this identification must be carried out in real time, so the proposed approach is more suitable for use in digital twins.

The accuracy of the digital twin is determined by the following:

- measurement error of object parameters (not considered in the article);

- the number of terms in functional expansions (following the Weierstrass theorem on the uniform approximation of a function by polynomials, a continuous function can be approximated arbitrarily accurately).

The approach considered in the article is not an alternative to artificial intelligence methods but rather a tool that expands the capabilities of artificial intelligence methods.

2.Language can be further improved.

We have tried to improve the English language in our article

Reviewer 3 Report

-        In Page 2 the authors mention: ” these modeling methods belong to the class of simulation models that do not take into account real physical and chemical processes, and therefore do not guarantee their correctness in the entire range of operating conditions”. While this statement in general is correct, it may happen that the underlying governing behavior of a system (and, in turn, the mathematical equations and physical laws that describe its evolution) may be unknown; thus, digital twin modelling may resort to such types of simulation models to approximate this unknown behavior. Please comment on this case as well.

-        One other aspect for resorting to robust mathematical models instead of black-box approaches is interpretability and explainability for subsequent decision-making. Authors should emphasize this aspect in the introduction of the paper.

-        In the definition of the system of equations (1) - (3), do the authors account for the noise in the acquired measurements. How would numerical differentiation be accurately computed when having noisy measurements? In addition, would equality in Eq. (23) hold in such a case?

-        How do you disambiguate between the impact of the internal system dynamics and the impact of the external energy flows, on the change of the system state?

-        Does the change of coordinates always impact the dimensionality of the original system state? How is the function r_x in Eq. (29) determined? Have you also quantified the tradeoff with accuracy after the change of the coordinate basis?

The authors are encouraged to perform additional proofreading throughout the paper. Several typos and grammatical errors are spotted. Examples:

-        Page 1, “which is significantly depend not only”

-        i.e. should be followed by comma: i.e.,

Author Response

Above all, we would like to thank you for reviewing our article and we appreciate your reasonable suggestions. All your suggestions have been carefully considered and worked out. The answers and explanation of our point of view are presented below:

  • In Page 2 the authors mention: ” these modeling methods belong to the class of simulation models that do not take into account real physical and chemical processes, and therefore do not guarantee their correctness in the entire range of operating conditions”. While this statement in general is correct, it may happen that the underlying governing behavior of a system (and, in turn, the mathematical equations and physical laws that describe its evolution) may be unknown; thus, digital twin modelling may resort to such types of simulation models to approximate this unknown behavior. Please comment on this case as well.

I agree with your statement. Moreover, it should be noted that new phenomena not yet studied by science are almost always characterized by uncertainty at the level of physical description. In this case, the only option to obtain a model is to use a simulation model. However, if we are dealing with technical systems for which a digital twin is being designed, then, as a rule, the physical and chemical processes in them are known.

The approach proposed in the article is valid for macroscopic systems consisting of a large number of randomly interacting particles; these macroscopic systems must satisfy the principles of thermodynamics.

  • One other aspect for resorting to robust mathematical models instead of black-box approaches is interpretability and explainability for subsequent decision-making. Authors should emphasize this aspect in the introduction of the paper.

Thank you for your recommendation. This aspect is added in the introduction.

  • In the definition of the system of equations (1) - (3), do the authors account for the noise in the acquired measurements. How would numerical differentiation be accurately computed when having noisy measurements? In addition, would equality in Eq. (23) hold in such a case?

The article considers the assumption that the noise level of the meters does not exceed a value that guarantees the possibility of approximating the data with smooth curves. The analytical expression of the approximating curve is differentiated.

The equality in Eq. (23) holds because the average data of the measured parameters are considered in equation (23).

  • How do you disambiguate between the impact of the internal system dynamics and the impact of the external energy flows, on the change of the system state?

It is a good question. The presented account of internal and external energy flows is the basis of the method of mathematical prototyping of energy processes. The influence of internal and external energy flows is unambiguously determined by the laws of conservation, and the principles of thermodynamics, including the kinetic theorem of modern non-equilibrium thermodynamics. Due to this, the correctness of the obtained mathematical models is guaranteed.

  • Does the change of coordinates always impact the dimensionality of the original system state? How is the function r_x in Eq. (29) determined? Have you also quantified the tradeoff with accuracy after the change of the coordinate basis?

The dimensionality of the state coordinates of the system is determined only by the objective properties of the object of study and does not depend on the chosen coordinate system (basis).

 rx(x‾,w,γ) is a transition function from one state coordinates to another. The transition to new coordinates is determined by the features of the use of the digital twin (testing, operation, forecasting, diagnosing, etc.). In this case, the accuracy of the models may change, but it is always possible to find a state coordinate system that provides the maximum accuracy of the model. In this article, the study of the influence of the choice of the coordinate system on the accuracy of the model is not considered.

  • The authors are encouraged to perform additional proofreading throughout the paper. Several typos and grammatical errors are spotted. Examples:...

We have tried to improve the English language in our article 

Round 2

Reviewer 3 Report

Authors have addressed my review comments raised in the previous review round in a satisfactory manner.